# Priority-based transformations of stimulus representation in visual working memory

Quan Wan [1]*, Jorge A. Menendez [2], Bradley R. Postle [1,3]

**1** Department of Psychology, University of Wisconsin–Madison, Madison, Wisconsin, United States of America, **2** Gatsby Computational Neuroscience Unit, University College London, London, United Kingdom, **3** Department of Psychiatry, University of Wisconsin–Madison, Madison, Wisconsin, United States of America

* qwan22@wisc.edu

**Data Availability Statement:** All processed human EEG data, code, network training sets and trained networks are available at https://osf.io/sgqvn/ (DOI 10.17605/OSF.IO/SGQVN) on Open Science Framework.

## Abstract

How does the brain prioritize among the contents of working memory (WM) to appropriately guide behavior? Previous work, employing inverted encoding modeling (IEM) of electroencephalography (EEG) and functional magnetic resonance imaging (fMRI) datasets, has shown that unprioritized memory items (UMI) are actively represented in the brain, but in a "flipped", or opposite, format compared to prioritized memory items (PMI). To acquire independent evidence for such a priority-based representational transformation, and to explore underlying mechanisms, we trained recurrent neural networks (RNNs) with a long short-term memory (LSTM) architecture to perform a 2-back WM task. Visualization of LSTM hidden layer activity using Principal Component Analysis (PCA) confirmed that stimulus representations undergo a representational transformation–consistent with a flip—while transitioning from the functional status of UMI to PMI. Demixed (d)PCA of the same data identified two representational trajectories, one each within a UMI subspace and a PMI subspace, both undergoing a reversal of stimulus coding axes. dPCA of data from an EEG dataset also provided evidence for priority-based transformations of the representational code, albeit with some differences. This type of transformation could allow for retention of unprioritized information in WM while preventing it from interfering with concurrent behavior. The results from this initial exploration suggest that the algorithmic details of how this transformation is carried out by RNNs, versus by the human brain, may differ.

## Author summary

How is information held in working memory (WM) but outside the current focus of attention? Motivated by previous neuroimaging studies, we trained recurrent neural networks (RNNs) to perform a 2-back WM task that entails shifts of an item's priority status. Dimensionality reduction of the resultant activity in the hidden layer of the RNNs allowed us to characterize how a stimulus item's representation follows a transformational trajectory through high-dimensional representational space as its priority status changes from memory probe to unprioritized to prioritized. This work illustrates the value of artificial

**Funding:** This work was supported by a grant from National Institutes of Health (http://www.nih.gov; grant no. R01-MH064498) awarded to BRP. The funders had no role in study design, data collection and analysis, decision to publish, or preparation of the manuscript.

**Competing interests:** The authors have declared that no competing interests exist.

neural networks for assessing and refining hypotheses about mechanisms for information processing in the brain.

## Introduction

The ability to flexibly select and prioritize among information held in working memory (WM) is critical for guiding behavior and thought. To do this successfully, the cognitive system must solve a fundamental computational problem of how to maintain information in a readily accessible state while also preventing it from interfering with ongoing behavior. The primary goal of the work presented here is to investigate how this might be accomplished. If two items are currently held in WM, one possible solution could be to encode the "unprioritized memory item" (UMI) into a pattern of synaptic weights [1,2], an "activity-silent" trace [3] that might be less likely to interfere with the currently active "prioritized memory item" (PMI). Although some previous neuroimaging studies have reported that the prioritization of one item held in WM leads to a decrease-to-baseline of the activity level of the UMI [4–6], whether an "activity-silent" mechanism may contribute functionally to WM remains a topic of vigorous debate [7–10]. In the present report, we evaluate an algorithmically different solution for prioritization: the *priority-based transformation* of the UMI into a representational format that, although active, is different from that of the PMI. Such a transformation could minimize the likelihood that the UMI interferes with ongoing behavior.

Experimental tasks used to study prioritization in WM necessarily include multiple steps, such that information not needed for the impending response (i.e., the UMI) might nevertheless be needed to guide a subsequent response. This is often done with retrocues, and two recent studies using a retrocuing procedure have provided evidence consistent with priority-based transformation. In one, van Loon and colleagues [11] acquired functional magnetic resonance imaging (fMRI) data while first presenting subjects two target images sequentially (e.g., first a flower then a cow), then indicating with a cue whether memory for the first or second presented image would be tested first. Had the cue been a "1", subjects would next see a test array of six flowers and indicate whether the target flower appeared in the test array, and finally a test array of six cows. On this trial, the target cow spent time as UMI, because the cue indicated that memory for the flower would be tested first. When van Loon et al. [11] applied multivariate pattern analysis (MVPA) to fMRI data from posterior ventral temporal lobe, they found that a decoder trained on trials when an item was a PMI performed statistically below chance when that item was a UMI. Furthermore, a representational dissimilarity analysis indicated that, within their set of 12 stimuli (four cows, four skates, four dressers), each item's high-dimensional representation in one state (e.g., as a PMI) was maximally different from its representation in the other state (i.e., as a UMI). Using a similar retrocuing procedure, Yu, Teng and Postle [12] found, with multivariate inverted encoding modeling (IEM) of fMRI data from early visual cortex, that the reconstructed orientation of a grating "flipped" when it was a UMI relative to a PMI (e.g., a 30˚ orientation reconstructed as 120˚ while a UMI). Furthermore, for data from the intraparietal sulcus (IPS), they observed that the IEM reconstruction of the location where an item had been presented also flipped when an item's priority status transitioned to UMI.

Shifts of priority are also characteristic of continuous-performance tasks, for which shifts of priority are dictated by task rules rather than by explicit cues. One example, which features prominently in the work presented here, is the 2-back WM task from Wan and colleagues [13] (Fig 1). Electroencephalography (EEG) signals were recorded while subjects viewed the serial

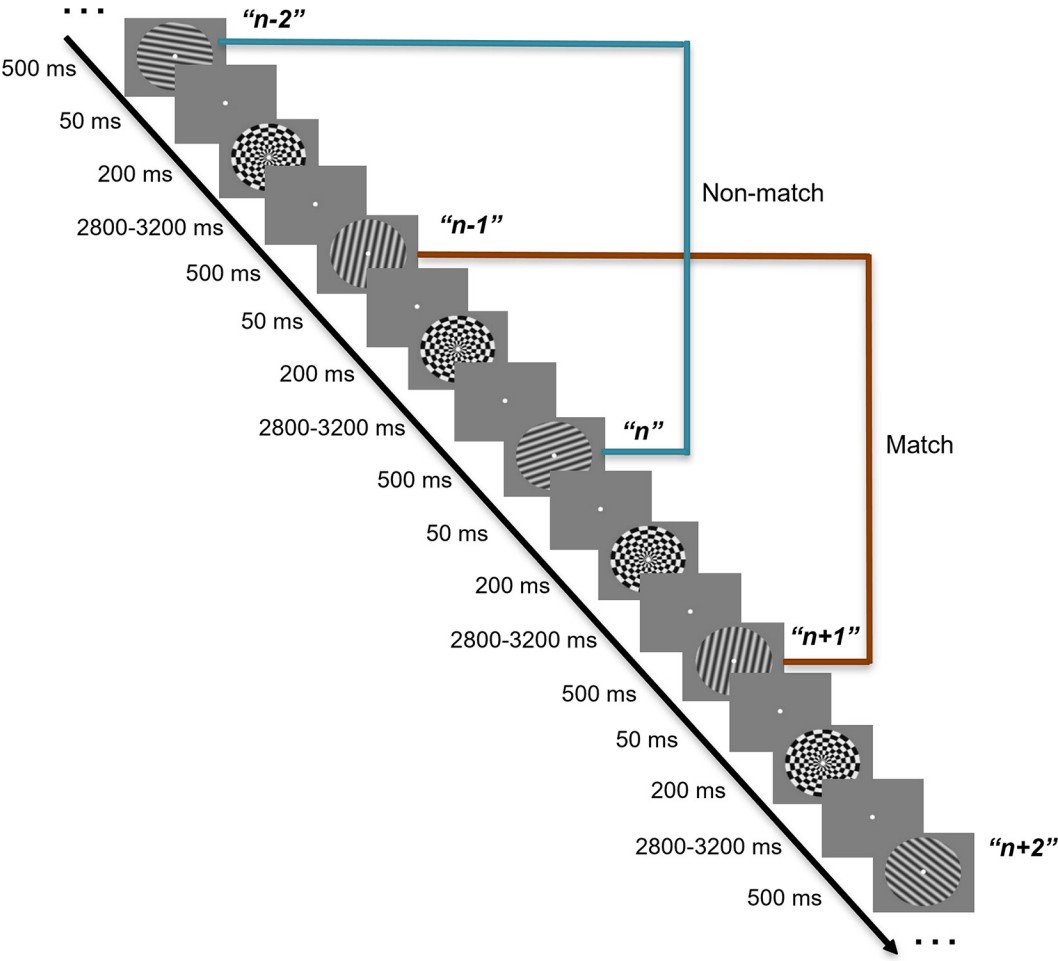

**Fig 1. 2-back task structure in the Wan et al. [13] EEG study.** The presentation of each stimulus is followed by a 50 ms blank screen, a 200 ms radial checkerboard mask, a variable delay from 2.8 to 3.2 s, and then the next stimulus was presented, upon which the match vs. non-match response is to be made.

presentation of oriented gratings and judged for each one whether it was a match or a non-match to the item that had appeared two positions previously in the series. This task entails a predictable transition through priority states for each item: When an item *n* is initially presented, it serves as probe to compare against the memory of item *n– 2*; after the *n*-to-*n- 2* decision is made, item *n* becomes a UMI while item *n– 1* is prioritized for the upcoming comparison with *n + 1*. Next, once the *n + 1-to-n– 1* comparison is completed, item *n* becomes a PMI for its impending comparison with item *n + 2*. To analyze the EEG data, an IEM was trained on the raw EEG voltages from a separate 1-item delayed-recognition task, and then tested on the delay periods separating *n* and *n + 1* and separating *n + 1* and *n + 2* (i.e., when item *n* assumed the status of UMI, then PMI). The results, reminiscent of van Loon et al. [11] and Yu, Teng and Postle [12], indicated that the IEM reconstruction of the UMI was "flipped" relative to the training data (Fig 2). The authors referred to this transition from PMI to UMI as "priority-based remapping" (rather than "recoding" or "code morphing"; c.f. [14]), reasoning that the IEM reconstruction of the UMI would fail if it were represented in a neural code different from the trained model.

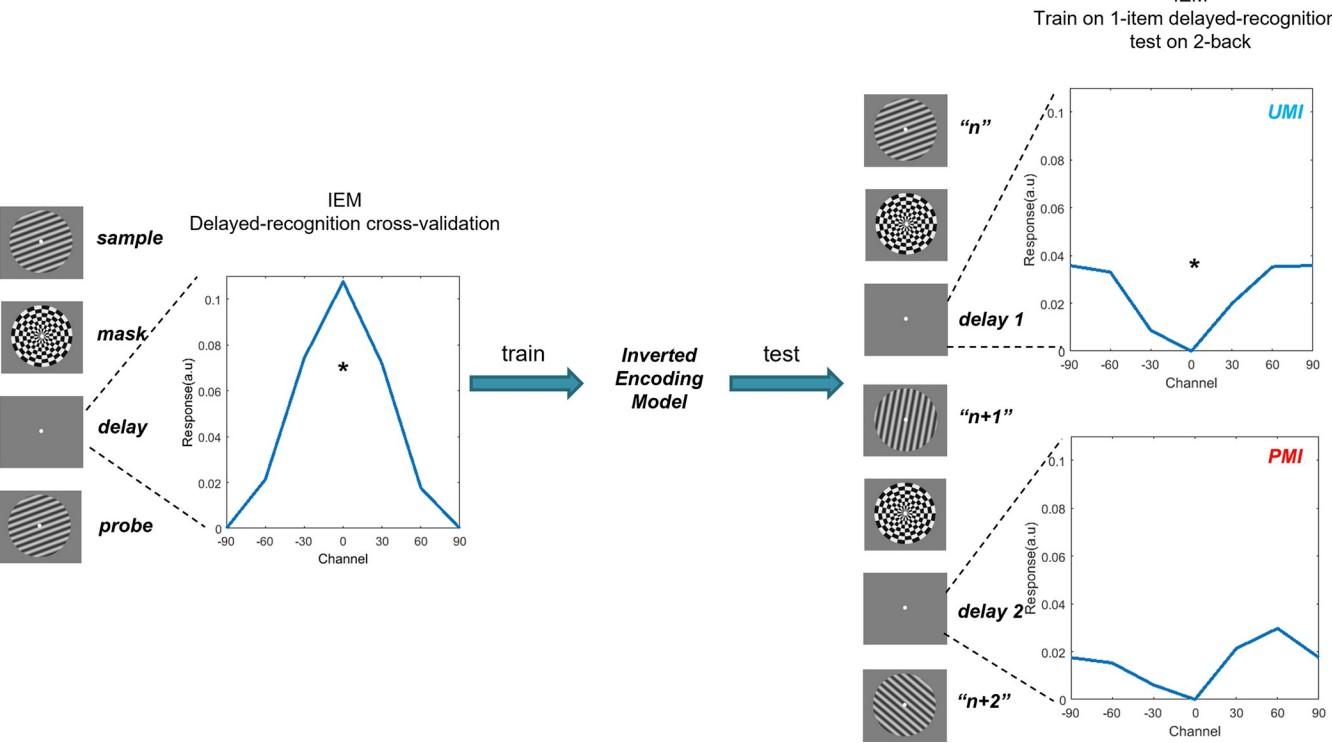

**Fig 2. IEM reconstruction of EEG recorded while subjects performed the 2-back task ($N$ = 42, combining data from the pilot study and the preregistered experiment from Wan et al. [13]).** In IEM, voltage from each EEG electrode is construed as a weighted sum of responses from six orientation channels (modelled by a half-wave-rectified sinusoid raised to the 6th power), each tuned to a specific stimulus orientation, comprising the basis set. Left panel: IEM reconstruction of the stimulus during the delay in a separate one-item delayed-recognition task. This model was used to reconstruct the stimulus in the 2-back task. Right panel: Concatenation of the item $n$ and item $n$ + 1 stimulus events to form a trial, across which $n$ transitions from probe to UMI to PMI in the 2-back. On the right are IEM reconstructions corresponding to the two 2 s windows centered in two 2.8 s post-mask ISIs before and after item $n$ + 1, respectively. "*" indicates $p$ < .01 (two-tailed $t$ test), FDR-corrected for multiple comparisons. As the figure shows, IEM reconstruction of stimulus $n$ is "flipped" relative to the training data (IEM reconstruction from delayed recognition) when it is a UMI, demonstrating priority-based remapping. (Reconstruction of the PMI was unsuccessful.) For delayed-recognition IEM reconstruction (940–1040 ms from stimulus onset), $t(41)$ = 4.12, $p$ < 0.001. For UMI reconstruction of 2-back (-2400 –-400 ms relative to $n$ + 1 onset), $t(41)$ = -3.02, $p$ = 0.009; for PMI reconstruction of 2-back (1150–3150 ms from $n$ + 1 onset), $t(41)$ = -1.60, $p$ = 0.116.

Two recently published computational models offer some insight into the empirical phenomena that we have described up to this point. One model, by Lorenc and colleagues [15], was designed to account for a similar flipped IEM reconstruction observed in an fMRI study using a retrocuing task. This approach was inspired by evidence from nonhuman primates (NHP) performing WM tasks, in which top-down signals from FEF were shown to alter several receptive field properties of neurons in extrastriate visual areas V4 and MT [16]. They created simulated data for training IEMs using the basis set that was employed for IEM reconstructions of empirical data, and subsequently created a test dataset where the basis function parameters for memory strength, gain, receptive field width, and receptive field centers were varied. When these parameters were fitted to experimental data, the best solution was a selective down-modulation of gain in feature-tuned sensory channels paired with a weakly excitatory top-down signal (i.e., memory strength). A second model, from Manohar and colleagues [17], simulated WM performance in a network comprised of hard-coded feature-selective units and a pool of freely conjunctive units that can form a plastic attractor to keep one item, a PMI, in a state of elevated activity. When attention shifted away from an item (making

it a UMI), it remained briefly encoded in a residual pattern of strengthened connections, and, under some conditions, inhibition from activity in other parts of the network produced an "inverted" representation of UMI. Although this model successfully reproduced other empirical findings using simulated data, such as the temporary reactivation of the UMI by a nonspecific pulse of excitation, it was not used to account for empirical neural data.

It is instructive to consider the two models reviewed above from the perspective of the framework of Marr and Poggio [18]: They address distinct *computational* problems–prioritization within WM [17] vs. removal from WM [15]–they propose different *algorithmic* solutions–inhibition via biased competition [17] vs. excitation paired with selective gain modulation [15]–yet they observe similar patterns of neural *implementation*–flipping. Of particular relevance for our interests here is that although the details of their algorithmic operations differ, both models are constrained to finding only one class of solution: changing the strength of attention. Importantly, neither allows for the alternative that we will test here, which is the transformation of an item's representational geometry.

Previous WM research has implicated representational transformation as a solution to a third computational problem for WM: the retention of information in the face of distraction (e.g., [14,19]). Our goal with the present work was to explore the possibility that the computational problem of prioritization in WM might also be solved algorithmically via representational transformation. To accomplish this we turned to artificial neural networks (ANNs), which have been playing an increasingly prominent role in providing mechanistic insights into, and generating novel hypotheses of, phenomena in cognition and neuroscience [20–24]. In the current work, we use recurrent neural networks (RNNs) with an LSTM architecture [25] to perform a 2-back WM task modeled on [13]. LSTMs can generate flexible behavior guided by long-range temporal dependencies, and can solve complex tasks such as speech recognition [26] and machine translation [27]. Moreover, LSTM might be a good model for WM tasks due to its gating-based architecture, reminiscent of the cortico-striatal mechanisms believed to gate information into and out of WM [28,29]. By comparing the stimulus representational schemes embedded in the EEG and RNN data, we hope to reveal whether humans and RNNs might employ similar algorithmic principles. Given that the RNNs are optimized to solve the 2-back task, we can also potentially use the RNN results to evaluate whether the algorithm that humans use reaches optimality.

Our approach was to train RNNs to perform the 2-back task, and then first use Principal Component Analysis (PCA) of the activity of the RNN's hidden layer to visualize its representational dynamics. This revealed a smooth rotational transformation of stimulus representations over the course of the trial. This trajectory provided novel, independent evidence that the transition of the functional role of an item from memory probe to UMI to PMI is accompanied by transformations of its representational format. However, because PCA does not allow for the isolation and quantification of variation attributable to specific task dimensions (of particular interest here, priority status and the match/nonmatch decision), we carried out two additional sets of analyses. First, we applied demixed Principal Component Analysis (dPCA; [30]) to the RNN data in order to identify distinct low-dimensional subspaces occupied by the neural representations of the UMI, the PMI, and the RNN's decision. We then quantified the temporal dynamics of these representations within the subspaces and the geometric relationships between the subspaces. Finally, we used this analysis of the RNN data to derive quantitative hypotheses with which to assess evidence that the EEG data from Wan et al. [13] may also show evidence of priority-based transformation. The results of these hypothesis tests provide novel insights about priority-based transformations of stimulus information that are carried out by the human brain.

## Methods

### Ethics statement

The experimental protocol for the Wan et al. [13] EEG study (the data from which was analyzed in this paper), along with the informed consent form, was approved by the University of Wisconsin–Madison Health Institutional Review Board (protocol no. 2016–0500). Prior to each experimental session, written informed consent was obtained by lab personnel listed on the IRB-approved protocol.

### Behavioral task

In each experimental block of the 2-back WM task, both human subjects ($N$ = 42) and RNNs ($N$ = 20) were serially presented a sequence of stimuli drawn from a closed set of six different identities (128-stimulus blocks for humans, 20-stimulus blocks for RNNs). The task was to indicate, for each stimulus, whether or not it matched the identity of the stimulus that had been presented 2 positions earlier in the series. Each EEG subject performed 4 blocks and each RNN performed 200 blocks.

### Recurrent neural network (RNN) model

**RNN architecture.** Twenty RNNs with an LSTM architecture were trained and simulated using the Python-based machine learning package PyTorch. Specifically, we used the default LSTM architecture in PyTorch with its default initializations. Initially, we trained 10 networks that consisted of 6 input neurons and 7 LSTM hidden units, which were linearly rectified and linearly read out to a single output neuron (Fig 3). We initially chose to use 7 units because this was the smallest number that solved the task with network solutions that were highly consistent across training instances (as evaluated by representational dynamics from the PCA visualization). Networks with other numbers of hidden units (up to 256) gave qualitatively similar results. Subsequently, we repeated the procedure with RNNs with 60 LSTM hidden units to match the dimensionality of our EEG data, and with an input structure simulating the orientation stimuli in the human 2-back task of Wan et al. [13].

**Stimuli.** For 7-hidden-unit networks, the identity of each stimulus presented to the network was denoted by an integer randomly generated between 1 and 6. The stimulus input took the form of a one-hot vector, with only the unit corresponding to the stimulus identity activated (e.g., [0, 0, 1, 0, 0, 0] for stimulus #3; we also explored RNNs trained on metrically varying input vectors following the basis function used to build IEMs in Wan et al. [13], and these yielded similar results, see Fig A in S1 Text). For 60-hidden-unit networks, to simulate the orientation stimuli, we instead employed 2 input units taking the vector [cos 2θ, sin 2θ], where θ denotes the orientation angle of each stimulus used in Wan et al. ([13]; 10˚, 40˚, 70˚, 100˚, 130˚, 160˚). We multiply the angle θ by 2 to reflect the circular structure of the oriented grating stimuli, which have a period of 180˚ (i.e., a $K$˚ stimulus is identical to a 180˚ + $K$˚ stimulus). The multiplication by 2 ensures that this is also true for the corresponding inputs: cos (2 * (180˚ + $K$˚)) = cos(2$K$˚) and sin (2 * (180˚ + $K$˚)) = sin (2$K$˚) (see Fig B in S1 Text for more details). To simulate the delay period in the human task, we installed 2 "delay" timesteps following the presentation of each stimulus (with an input of [0, 0, 0, 0, 0, 0]; no delay timesteps after the last stimulus in the sequence). A "stimulus event" consisted of the presentation of stimulus $n$ and its following two delay timesteps. To evaluate the UMI-to-PMI representational transition of stimulus $n$, we refer to the concatenation of each two consecutive "stimulus events" as a "trial".

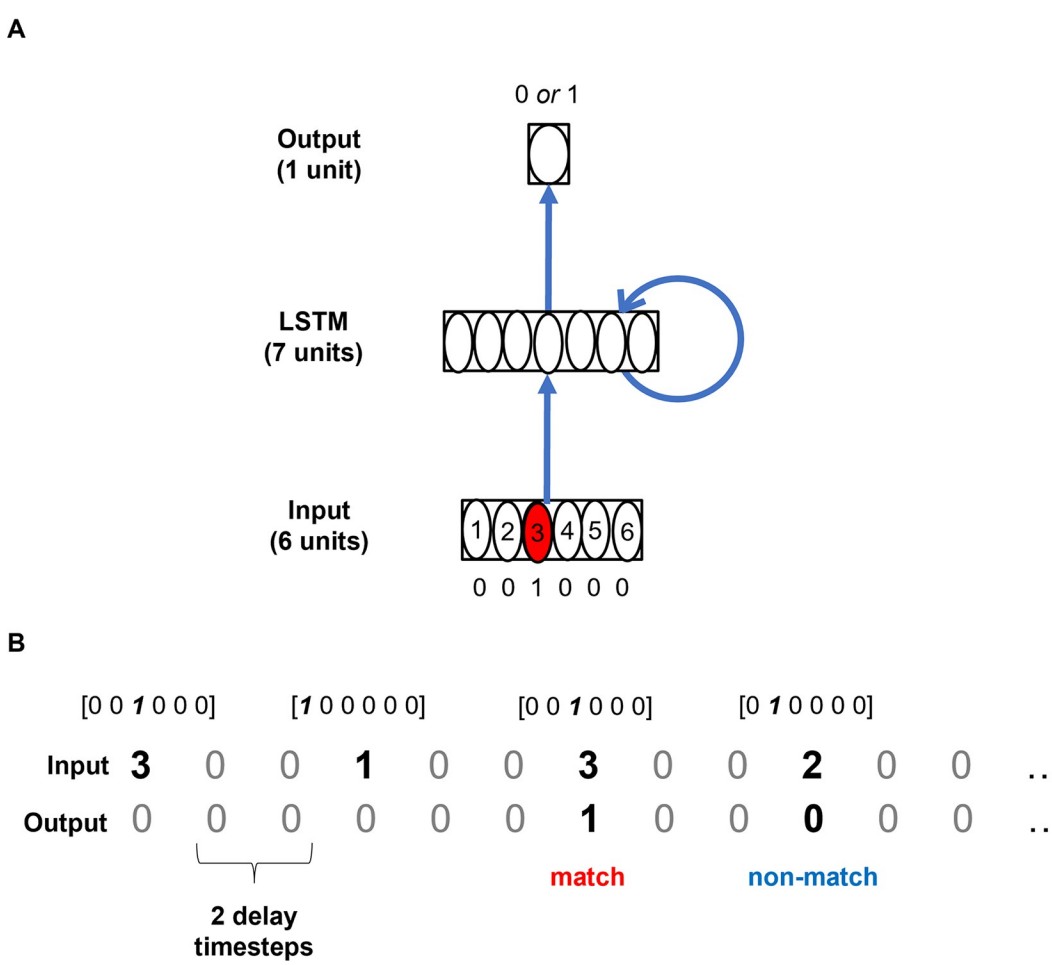

**Fig 3. RNN model architecture.** Shown is the architecture of the 7-hidden-unit RNNs. (A) One-hot vectors corresponding to each of the 6 stimulus types are fed into the input layer, which projects to an LSTM layer with 7 hidden units. This hidden layer in turn projects to an output unit with a binary target activation (0 = non-match, 1 = match). (B) Example input and target output sequences. Two delay timesteps were installed after each stimulus presentation timestep to emulate the delay period in the 2-back EEG task. 60-hidden-unit RNNs have the same architecture except that they have 60 LSTM hidden units, and two input units that take a vector [cos 2θ, sin 2θ] (θ denoting the angle of grating orientations used in Wan et al. [13]) instead of a one-hot vector.

A scalar output was read out from the LSTM internal state by linearly rectifying the hidden state and then applying a linear layer. The network was then trained to output 1 (match) or 0 (non-match) during stimulus presentation depending on whether the presented stimulus (*n*) matched the stimulus presented two stimulus events back (*n—2*). Each stimulus sequence comprised 18 "trials" (as defined above–note that because no delay period followed stimulus #20; the last "trial" contains stimuli #18 and #19), and only 16 trials were analyzed (because the first two stimulus events had no target outputs: not enough stimuli preceded them for there to be a match/non-match decision). We generated 200 random stimulus sequences for training the RNNs and 200 random sequences for testing the trained networks. Because the human 2-back task had a ratio of 1:2 between match and non-match trials, we generated random sequences that satisfied the criterion that each sequence had to contain at least 5 match trials. The outcome was that training sequences had an average of 5.55 match trials (*SD* = 0.78) and testing sequences an average of 5.46 match trials (*SD* = 0.70).

**RNN training and testing.** The internal state of the RNNs was initialized to 0, and weights and biases were initialized to random values, following the standard initialization of the PyTorch LSTM implementation. The 7-hidden-unit RNNs were trained using the Adam stochastic gradient descent (SGD) algorithm for 5000 iterations ([31]; learning rate = $10^{-3}$). In each iteration, a batch of 20 sequences was randomly selected (with replacement) from the 200 training sequences. The loss function minimized was the mean squared error between output activity and target output across all timesteps.

We observed that after 5000 iterations of training, most RNNs had excellent performance on the training data. We therefore stopped training at this point and evaluated each RNN on an independently sampled set of 200 test stimulus sequences to assess generalization to arbitrary stimulus sequences. The network's performance accuracy was calculated as the percentage of trials (across all 200 sequences in the test set) on which the network made a correct response, where a response was deemed correct if the absolute difference between the activation of the output neuron and the target output was smaller than 0.5. We set a criterion level of performance accuracy of 99.5% for the networks. A total of 12 7-hidden-unit networks were trained, 2 of which were discarded due to below-criterion performance, leaving 10 RNN's for our analysis. All RNNs trained had the same architecture, hyperparameters and training/testing sequences. The only thing that differs across these 10 networks is the random initialization of the RNN weights prior to training. For subsequent analyses, the activity timeseries of the LSTM hidden layer units from all 3200 trials (16 trials x 200 sequences) in the training data set were used.

After analysis of the 10 successfully trained 7-hidden-unit networks, we repeated these training procedures and trained 10 RNNs with 60 units in the LSTM layer (batch size = 20, learning rate = $10^{-3}$, 1500 iterations), so as to generate RNN data matching the dimensionality of our EEG data sets.

**PCA visualization of the LSTM layer activity.** We extracted from each network the activity of the 7 hidden units in the LSTM layer from all 200 training sequences and used Principal Component Analysis (PCA; implemented using Python's 'scikit-learn' library) to project these 7-dimensional activity patterns onto the top two dimensions accounting for the most variance across all training sequences and timesteps. We then visualized each stimulus *n*'s transition from probe to UMI to PMI within this subspace by plotting the dimensionality-reduced activity across the 9-timestep time course of a trial. These 9 timesteps comprised the presentations of stimulus *n*, *n + 1*, *n + 2* and the delay timesteps that followed each (i.e., *delay 1:1 and delay 1:2; delay 2:1 and delay 2:2; and delay 3:1 and delay 3:2*; Fig 4, "unlabeled" column). Note that, once a decision has been made about item *n + 2*, item *n* is no longer relevant for the task, so the *delay 3:1* and *delay 3:2* timesteps illustrate the evolution of the representational structure of *n* after it has presumably been "dropped from WM".

To see how the representation of stimulus *n* evolves as it transitions from being a UMI to a PMI, we colored the activity patterns according to the identity of stimulus *n* (Fig 4, "stimulus" column). As explained in the Introduction, the memory of stimulus *n* is a UMI during the delay period after the presentation of stimulus *n* (i.e., during *delay 1:1 and delay 1:2*; because it is not needed for the upcoming *n– 1*-to-*n + 1* comparison), and then becomes a PMI during the delay period after the presentation of stimulus *n + 1* (i.e., during *delay 2:1 and delay 2:2*; in preparation for the imminent comparison with *n + 2*). We focused on the *delay 1:2* and *delay 2:2* timesteps (highlighted by blue and red squares) to characterize the UMI-to-PMI representational transformation.

To visualize the representation of the RNN's decision, we re-plotted the same activity patterns but colored them according to the correct response ("match" or "non-match") to the *n*-to-*n + 2* comparison when *n + 2* was presented (Fig 4, "decision" column; *n*-to-*n + 2*

comparison timestep highlighted by yellow square). Note that, by construction, the RNN's actual response is the correct response in at least 99.5% of trials, so this coloring can be effectively thought of as the RNN's true response in each of these trials.

## WM-specific dimensionality reduction via dPCA

Demixed Principal Component Analysis (dPCA; [30]) was employed to identify dimensions of RNN and EEG activity relevant to the stimulus representation in WM. Traditional PCA identifies dimensions that maximize the total variance of the recorded activity patterns across all task variables, such as time, stimulus, and decision. Demixed PCA, on the other hand, identifies dimensions of activity that contain variability specific to individual task variables. Given a task variable of interest (e.g., stimulus identity), the dPCA algorithm groups recorded activity patterns according to this variable and then extracts dimensions that maximize variance *across* groups (e.g., activity patterns evoked by different stimuli) while also minimizing variance *within* groups (e.g., activity patterns evoked by the same stimulus, but at different points in time and with different decisions). Here, we used this method to identify dimensions of activity that were strongly modulated by the identity of the UMI or PMI during the delay period.

To extract the demixed Principal Components (dPCs) of UMI-related variance, we minimize the following loss function:

$$V^{UMI}, W^{UMI} = \arg \min_{V,W} \sum_{s,t} ||(\bar{x}^s - \bar{x}) - VW^T(x_t^s - \bar{x})||^2$$

where $x_t^s$ is the neural activity at time $t$ averaged over all trials in which stimulus $s$ ($s$ being one of the 6 stimuli) was the UMI (trial averaging was necessary to average away noise), $\bar{x}^s = \frac{1}{T}\sum_{t=1}^{T} x_t^s$ is its mean over time, and $\bar{x}$ is the global mean over all trials and timepoints. This least squares optimization problem is called reduced-rank regression, and admits a closed-form solution [30]. This objective seeks to capture fluctuations in activity, $\bar{x}^s - \bar{x}$, arising from changes in the UMI stimulus and independent of time, as we expect the WM representation to stay stable over the late delay period. We refer to the columns of $W^{UMI}$ as the *UMI dPCs*, and call the subspace spanned by the columns of $V^{UMI}$ the *UMI subspace*. We similarly extracted *PMI dPCs*, $W^{PMI}$, and a *PMI subspace*, $V^{PMI}$, by exactly repeating the above operation but with the index $s$ now indexing the PMI stimulus rather than the UMI stimulus.

In order to extract dimensions of activity specific to WM, we sought to restrict the above optimization to activity patterns during the late delay period. For the RNN's, this led us to utilize a single timepoint: the second timestep of the delay period (i.e., $t$ only takes on a single index; cf. *delay 1:2* (for UMI), *delay 2:2* (for PMI) in Fig 4). For the EEG data, we used timepoints from the second half of the delay: $t \in$[-1400ms, 0ms] (for UMI) and $t \in$[2150ms, 3550ms] (for PMI) relative to stimulus *n + 1* onset.

For the purposes of visualization we extracted only two dPCs (i.e. $V$ and $W$ each have two columns only), so as to obtain two-dimensional projections, $z_t^s$, of the neural activity. These projections were computed using the dPCs ($W^{UMI}$ or $W^{PMI}$) as follows,

$$z_t^s = W^T(x_t^s - \bar{x})$$

It is these two-dimensional vectors that are plotted in Fig 5 using the simulated RNN data ($x_t^s$ is the 7- or 60- dimensional internal LSTM state vector) and in Fig 6 using the EEG data ($x_t^s$ is the 60-dimensional vector of signals recorded at each EEG channel).

For estimating the geometric relationships between stimulus and decision subspaces (Fig 7), we estimated decision dPCs, $W^{dec}$, and a decision subspace, $V^{dec}$, by capturing variability

arising from changes in the subject's decision, $x_t^{s,d} - \bar{x}_t^s$, as follows,

$$V^{dec}, W^{dec} = \arg \min_{V,W} \sum_{s,t} ||(x_t^{s,d} - \bar{x}_t^s) - VW^T(x_t^{s,d} - \bar{x})||^2$$

where $x_t^{s,d}$ is the is the neural activity at time $t$ averaged over all trials in which stimulus $s$ was the probe and response $d$ ("match" or "non-match") was the decision made by the subject (or RNN), $\bar{x}_t^s = \frac{1}{2}\left(x_t^{s,"\text{match}"} + x_t^{s,"\text{non-match}"}\right)$ is its mean over the two decisions, and $\bar{x}$ is again the global mean over all trials and timepoints. We again used two dPCs, in accordance with previous analyses of WM subspaces [32]. In this case, we only considered timepoints during the decision time period: $t \in [200\text{ms}, 700\text{ms}]$ relative to stimulus onset for EEG and $t$ = stimulus presentation timestep for RNN. See "UMI/PMI/decision subspace analysis" section below on how the relationships between the different subspaces ($V^{UMI}$, $V^{PMI}$, and $V^{dec}$) were then quantified.

Percent variance explained calculations were performed as follows. Percent global variance explained by the $i$th dPC, $w_i$ (i.e. the $i$th row of the decoder matrix $W$), was calculated using the corresponding column $v_i$ from the encoder matrix $V$ by

$$1 - \frac{\sum_{s,t}||(x_t^s - \bar{x}) - v_i w_i^T(x_t^s - \bar{x})||^2}{\sum_{s,t}||(x_t^s - \bar{x})||^2}$$

The percent *stimulus* variance explained was defined as

$$1 - \frac{\sum_s||(\bar{x}^s - \bar{x}) - v_i w_i^T(\bar{x}^s - \bar{x})||^2}{\sum_s||(\bar{x}^s - \bar{x})||^2}$$

## Characterizing the dynamics of the UMI-to-PMI transformation

To characterize the continuous dynamics of the UMI-to-PMI transformation in stimulus-relevant dimensions, we quantified the evolving geometry of the stimulus representation visualized in Figs 5 and 6 by fitting a scalar transform, $k$, that minimizes the squared difference between the stimulus representation at a given timepoint and the transformed early-delay UMI representation,

$$\hat{k}_t = \arg \min_k \sum_s ||W^T(x_t^s - \bar{x}_t) - kW^T(\bar{x}^s - \bar{x})||^2$$

where $\bar{x}^s$ here is our estimate of the UMI, estimated by averaging activity patterns, $x_t^s$, over all timesteps $t$ during the first half of the first delay (*delay 1:1* for RNN and -2800 to -1400 ms relative to stimulus $n + 1$ onset for EEG). The index $s$, which refers to the stimulus presented prior to this delay, therefore corresponds to the identity of the UMI stimulus. In Figs 5B, 5D and 6B we plot these best-fitting scalars as a function of time over a whole trial, as the activity transitions from representing the stimulus as a UMI to representing it as a PMI. To isolate structure within the WM-relevant subspaces, we fit this transformation to low-dimensional projections through the UMI dPCs, $W^{UMI}$, or PMI dPCs, $W^{PMI}$.

Note that before computing these projections we center the activity vectors by subtracting their mean at the corresponding timepoint, $\bar{x}_t = \frac{1}{S}\sum_{s=1,...,S} x_t^s$. This is because we are specifically interested in how the representational format of the stimulus in WM changes over time, rather than changes in the absolute encoding of these stimuli. This analysis only sought to capture how the relationships between the different stimulus representations change over time.

## UMI/PMI/decision subspace analysis

To quantify the relationship between UMI, PMI and decision subspaces calculated from equations above, we used a metric developed by Panichello and Buschman [32]. This metric measures the alignment between corresponding pairs of dPC encoding vectors as follows:

$$\text{UMI} - \text{PMI subspace alignment} = |v_1^{UMI} \cdot v_1^{PMI}||v_2^{UMI} \cdot v_2^{PMI}|$$

$$\text{UMI} - \text{decision subspace alignment} = |v_1^{UMI} \cdot v_1^{dec}||v_2^{UMI} \cdot v_2^{dec}|$$

$$\text{PMI} - \text{decision subspace alignment} = |v_1^{PMI} \cdot v_1^{dec}||v_2^{PMI} \cdot v_2^{dec}|$$

where the dot denotes the Euclidean dot product, and the bars denote absolute value. Here, $v_1^{UMI}, v_2^{UMI}$ are the 1st and 2nd UMI dPC encoding vectors, i.e., the two columns of the matrix $V^{UMI}$. The analogous definition holds for the PMI and decision dPCs: $v_1^{PMI}, v_2^{PMI}$ are the columns of $V^{PMI}$; $v_1^{dec}, v_2^{dec}$ are the two columns of $V^{dec}$. Note that under the standard dPCA formulation used by Kobak et al. [30] and used here, the encoding vectors are all norm 1. These dot products can therefore be interpreted as cosines of angles between the pairs of vectors, and the subspace alignment metric can be interpreted as a product of two cosines.

To turn this metric into an angle, we took the inverse cosine of each alignment metric in the three equations above. These are the angles plotted in Fig 7.

## EEG dataset

60-channel EEG data were acquired and preprocessed as per procedures described in Wan et al. [13]. Raw EEG voltages were used for all analyses. Because data from the pilot and replication experiments from Wan et al. [13] yielded very similar IEM reconstruction results, they were combined to yield a dataset of 42 subjects. As is the case with the RNN data, after excluding the first two stimuli from each block there were 126 stimulus events and hence 125 trials per block. Each stimulus event (stimulus presentation followed by a delay) lasted 3550 ms. A third of the trials in each block were 'match' trials and the other two thirds were 'non-match' trials. EEG data from all trials (both correct and incorrect) were included in the analyses. For each stimulus $n$, during the delay period after its onset, stimulus $n- 1$ had the status of PMI and $n$ had the status of UMI.

# Results

## Behavioral results of EEG study

Mean accuracy was 86.1% ($SD$ = 5.6%), mean $d'$ was 2.40 ($SD$ = 0.65), and mean response time was 0.82 s ($SD$ = 0.18 s).

## Visualizing LSTM activity using PCA

PCA was carried out on the 7D LSTM hidden layer activity from the training data, and the resultant dimension-reduced activity from all 3200 trials projected onto the 2D-space constructed by the first 2 principal components (Fig 4, "unlabeled" column). This revealed that representations tended to cluster into band-like manifolds that appeared to rotate over the course of the trial (i.e., from timestep $n$ to timestep $n + 2$). Next, to get a sense of the stimulus representational structure and how it evolves over time, we colored the data points for each trial according to the identity of stimulus $n$ (Fig 4, "stimulus" column). This revealed that, across trials, stimulus representations were organized into stimulus-specific "stripes" that at

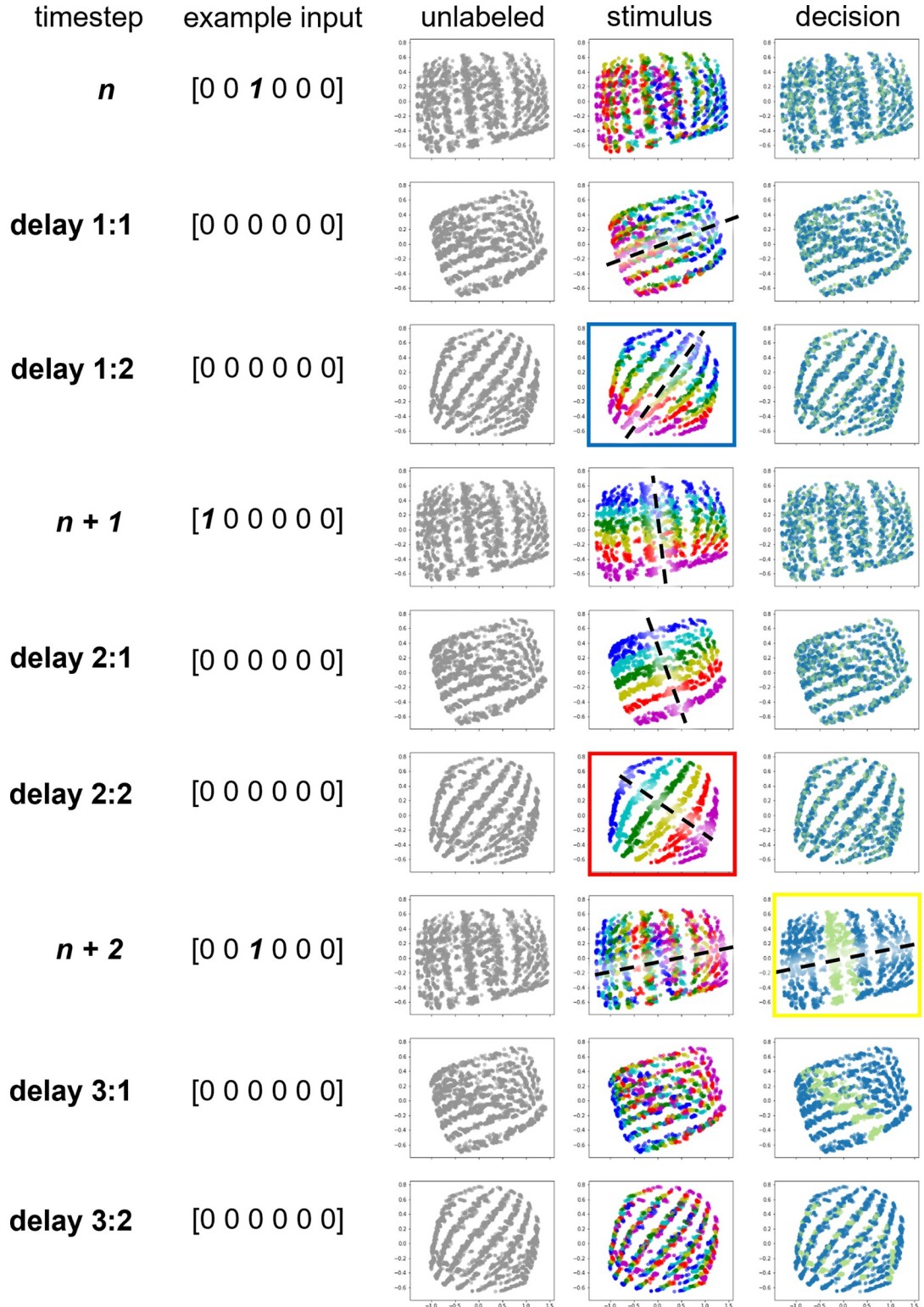

**Fig 4. PCA visualization of LSTM hidden layer activity of an example 7-hidden-unit network (#7).** Shown is a 9-timestep time course of the 2-back task, running from stimulus *n* to *delay 3:2*. Column 1 and 2: timestep labels and example input vectors. Column 3: Time course of dimensionality-reduced LSTM hidden layer activity. Each dot in the figures indicates the unit activity from a single trial. Column 4: Same as Column 3 but now each color corresponds to one of the six stimulus types, indicating stimulus *n*'s identity, and the black dashed lines illustrate the "schematic" stimulus coding axis. Blue and red squares

highlight the two delay timesteps used to identify UMI and PMI dPCs, respectively. Column 5: Same as Column 4 except that the colors now correspond to stimulus *n*'s status for the *n*-to-*n* + 2 comparison that occurs at timestep *n* + 2 (green: match trials, blue: nonmatch trials). Black dashed line at timestep *n* + 2 (yellow square) illustrates the decision-coding axis. As can be seen in Column 5, the stimulus coding axis rotates counterclockwise (in the image plane) over time such that it becomes "perpendicular" to the decision axis at timestep *n* + 1 and aligns with it at timestep *n* + 2. Percent variance explained: PC1–72.2%, PC2–15.7%.

some timesteps cut across the band-like manifolds (*delay 1:1* and *delay 1:2*), and at others were perfectly overlaid on them (*delay 2:1* and *delay 2:2*). These "stripes" thus defined a stimulus-coding axis. (That is, a stimulus's identity can be read out based on its location along this axis. A schematic illustration of this axis is superimposed on some of the timesteps from Fig 4, "stimulus" column, with a black dashed line.) It is noteworthy that, at timestep *n* + 2, the configuration of individual trials is different than at timestep *n*. This reflects that fact that items serve different functions at these two timesteps–*probe* at timestep *n* and *memorandum* at timestep *n* + 2. Indeed, if one were to re-color timestep *n* + 2 according to stimulus *n* + 2's identity, this frame would be identical to the configuration of stimulus *n* at timestep *n*, which means that *n* + 2 and *n* are in opposite locations in PCA space (e.g., in Fig 4, the azure-colored stimulus trials occupying the right side of PCA space at timestep *n* are on the left side of the space at timestep *n* + 2).

Finally, to get a sense of how the RNN's decision was represented, we colored each data point according to whether or not the correct response at the end of the trial was "match" (i.e., at timestep *n* + 2; Fig 4, "decision" column). This revealed that, when *n* is compared with *n* + 2, activity in trials requiring a "match" response converged onto the two central bands, whereas activity in non-match trials converged to the flanking bands. This organization thus defined a decision-coding axis, in that the correct response at a given trial can be read out based on the location of the RNN's internal state along this axis. A schematic illustration of this axis is superimposed on timestep *n* + 2 from Fig 4, ("decision" column, with a black dashed line).

Over the course of a trial, *n*'s stimulus-specific axis appeared to rotate counterclockwise (in the PCA plane) as it transitioned from UMI (during *delay 1:1 and delay 1:2*) to PMI (during *delay 2:1 and delay 2:2*). This likely reflects, in part, transitions between the functional roles of probe (timestep *n*), then UMI, then PMI. Thus, we can hypothesize the following functional account of the representational trajectory through a trial of, say, an azure-colored stimulus from Fig 4. At timestep *n*, its representational structure puts it on one of the central bands if it matches item *n*– *2* (and therefore elicits an output of [1]), or on a band to the right of center if it does not match item *n*– *2*. These two locations are separated along the decision-coding axis. Next, as it acquires the functional status of UMI, it transitions to a configuration that is not compatible with decision-making, as evidenced by the fact that every azure stimulus is located along a "stripe" that is parallel to the decision-coding axis at timestep *n* + *1* (stated another way, the stimulus-coding axis at timestep *n* + *1* is orthogonal to the decision-coding axis). During *delay 2:1* and *delay 2:2* the item's representation continues to rotate in the same counterclockwise direction on a trajectory that brings it back into alignment with the decision axis, but now on the "opposite side" of the PCA space, reflecting the fact that it is a PMI. (I.e., for azure items, probes cluster on the right side of PCA space, PMIs on the left side.) At timestep *n* + 2, the band occupied by this item will depend on its match/nonmatch status. From this we can further hypothesize that the function of this rotational trajectory might be to prevent the remembered representation of *n* from influencing the *n*– *1* versus *n* + *1* decision (at timestep *n* + *1*).

Whatever the intuitive appeal of these hypotheses, PCA is not well suited to reveal the structure most relevant for representing a given task variable (e.g., UMI/PMI status, decision, . . .), because PCA is completely agnostic about which task variables the neural activity depends on. We therefore next sought to more directly visualize the structure of the UMI, PMI, and decision representations by incorporating these task-relevant variables into our dimensionality reduction method.

### Visualizing LSTM representations using dPCA

Unlike PCA, which attempts to capture all variability across all time and all trials, dPCA seeks to capture variability dependent on specific task variables. By applying this dimensionality reduction method to neural activity during the delay period–during which the stimulus is held in memory–we can identify the dimensions most relevant to the representation of the stimulus in WM. By projecting the timeseries data into the subspaces spanned by these dimensions, we can visualize the temporal evolution of the geometry of the stimulus representation. This would allow us to test quantitatively the hypothesis that, for a given item $n$, its representational format while it is a UMI is transformed into a representational format that, although active, is different from that of the PMI.

**RNN with 7 LSTM units.** We applied dPCA to the 7D data from the RNNs to identify the top two UMI-selective dPCs (at the *delay 1:2* timestep) and the top two PMI-selective dPCs (at the *delay 2:2* timestep). The first two dPCs of the UMI subspace accounted for 97.4% of the total stimulus variance of the trial-averaged data. The first two dPCs of the PMI subspace accounted for 99.8% of the total stimulus variance (see Table A in S1 Text for additional information). Comparison of the trial-averaged population activity during the first delay period (*delay 1:2*) and second delay period (*delay 2:2*) reveals that, the way in which the stimulus is represented changes over time as it transitions from an unprioritized (UMI, in the first delay period) to a prioritized state (PMI, in the second delay period). The stimulus can be read out at both of these timepoints, but the relationship between stimulus and population activity is reversed (e.g., in Fig 5A, top row, the ordering along the 1$^{st}$ dPC at *delay 1:2* is *orange-yellow-purple-pink-teal-green*, whereas at *delay 2:2* it is *green-teal-pink-purple-yellow-orange*). This is true regardless of whether we project this representation through the UMI dPCs (Fig 5A, top row) or the PMI dPCs (Fig 5A, bottom row). Iteratively projecting trial-averaged activity from each timestep onto these two dPC subspaces suggested that the evolution of stimulus representational format across the trial is such that its projection onto the 1$^{st}$ dPC of the PMI–the axis that is critical for readout of the memory item against which the impending probe is to be compared at timestep $n + 2$—is minimal at timestep $n + 1$. (Note that this corresponds to the 0-crossing of the scalar transform, as described in the next paragraph.)

To quantify these dynamic changes in the stimulus representation across the various stages in the trial, we fit a scalar transformation from the trial-averages at timestep *delay 1:1* to the trial-averages at every other timestep (see Methods). The value of this best-fitting scalar transform for each timestep is plotted in Fig 5B. Relative to the UMI subspace (i.e., dPCA on timestep *delay 1:2*), an item's representational format was relatively stable (i.e., unchanging) for the first half of the trial, with the scalar transform close to 1.0, then, after timestep $n + 1$, shifted to a steady rate of transformation for the remainder of the trial, with the 0-crossing of the scalar transform (indicating the reversal of the stimulus-activity mapping) occurring at timestep *delay 2:1*. Relative to the PMI subspace, the representational format began contracting during *delay 1*, flipped just before timestep $n + 1$, and steadily expanding through *delay 2*. Together, these results confirm that an item's representational transformation across the trial proceeds at a relatively steady rate (consistent with the smooth rotation observed with the PCA (Fig 4)).

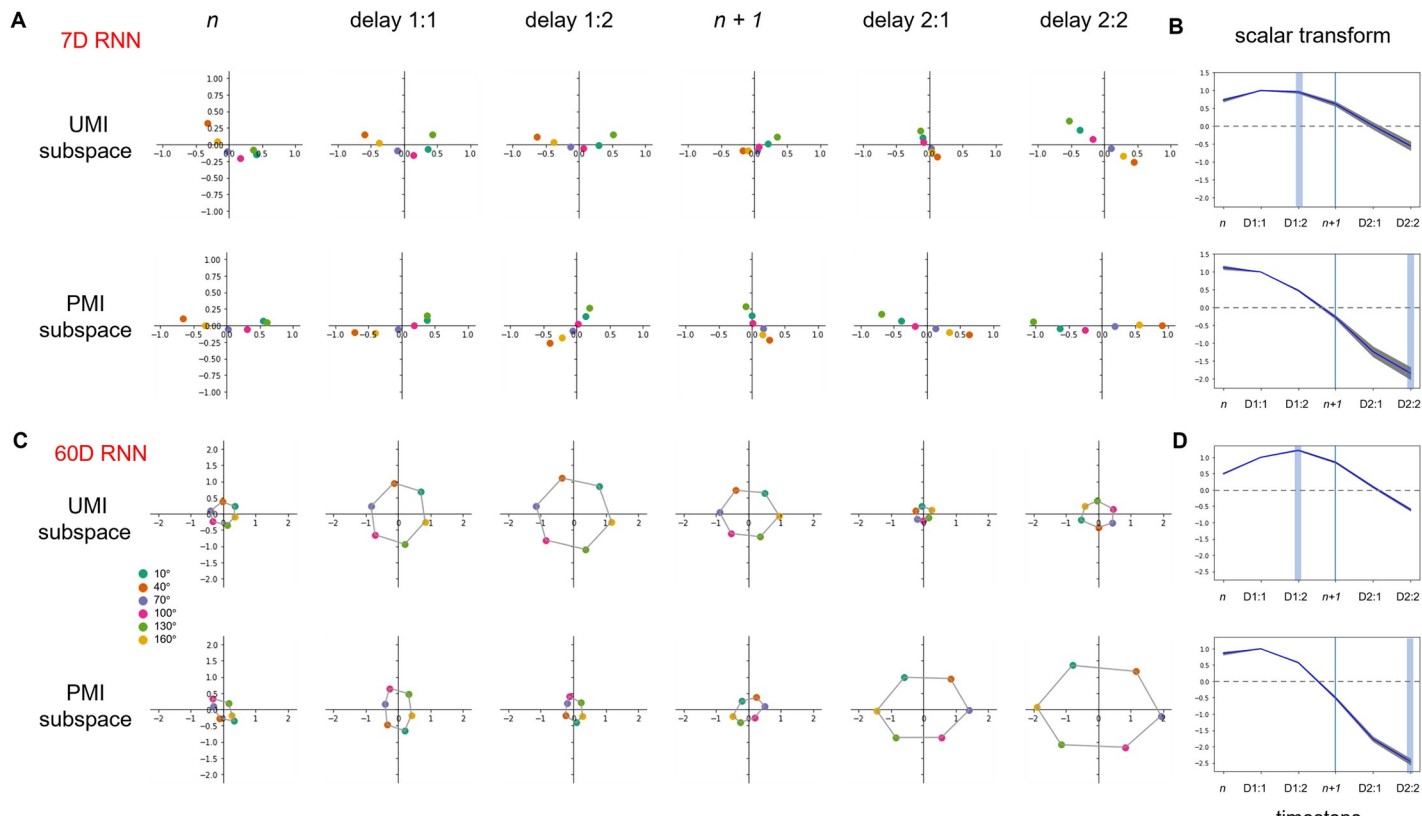

**Fig 5. Stimulus trial-averages of RNN hidden layer activity projected into UMI and PMI subspaces over the course of a trial (stimulus *n* to *delay 2:2*).** (A) Results from an example 7-hidden-unit network. Dot color indicates stimulus *n*'s identity. (B) Time course of scalar transform over the course of a trial, averaged across 10 networks. Blue vertical line indicates the timestep when stimulus *n + 1* is presented. Light blue shading shows the timesteps that were used to identify the dPCs. The gray shading around the curve indicates standard error of the mean. The gray dashed lines indicate a scalar transform of 0; the stimulus representational format is reversed after crossing this line. (C, D) Same as (A, B) but for the 60-hidden-unit RNNs. In C, data points of adjacent stimulus orientation angles are connected by a gray line.

**RNN with 60 LSTM units and a circular stimulus set.** Although the results from the 7D RNN data produced quantitative predictions about the priority-based transformation of information held in WM, their direct applicability to the EEG data from Wan et al. [13] would be complicated by two factors. First, there would be a difference in dimensionality between the two datasets (7D for the RNN, 60D [corresponding to 60 channels] for the EEG). Second, whereas the six stimuli used to train the RNNs with 7 LSTM units were unrelated to each other, the six stimuli used in Wan et al. [13] were orientations equally spaced within the circular range of 180˚. Therefore, our next step was to repeat the procedure described up to this point, but with 10 RNNs with 60 LSTM units each, trained on six stimuli drawn from a circular space. Results with the resultant 60D data would constitute the hypotheses that we would then test with the EEG data from Wan et al. [13].

To incorporate circular stimuli into our RNN model, we used 2D inputs taking the value [cos 2θ, sin 2θ], matching the periodicity of the oriented grating stimuli used in the task (where a 0˚ stimulus is equivalent to a 180˚ stimulus). We then constructed the six stimulus inputs by simply plugging in the six stimulus angles used in the EEG experiment. With these modifications, we trained 10 LSTMs with 60 hidden units to perform the 2-back task at > 99.5% correct. We then applied dPCA to the resultant 60-dimensional data from these RNNs. In this case, we found the UMI and PMI representations to have circular structure (Fig

5C), spreading across both dPCs rather than just one as we saw in the previous simulations. More concretely, the first two dPCs of the UMI subspace accounted for 98.7% of the total stimulus variance of the trial-averaged data. The first two dPCs of the PMI subspace accounted for 99.0% of the total stimulus variance (see Table A in S1 Text for additional information).

Based on our analysis of these RNN's, we derived two predictions that we next sought to test in the EEG data from Wan et al. [13]:

- *UMI-to-PMI representational reversal*: as in the 7D RNN, the stimulus representation reverses as it transitions from being unprioritized (*delay 1:2*) to prioritized (*delay 2:2*). That is, when the stimulus is a PMI (*delay 2:2*), the colored points in Fig 5C are flipped with respect to the *x*- and *y*-axes compared to when the stimulus is a UMI (*delay 1:2*). This can be quantified by characterizing the stimulus representation at each timestep as some scalar transformation of the representation early in the first delay period (*delay 1:1*). This scalar transform remains positive and near 1.0 during the entirety of the first delay period and then gradually decreases in value and reversing sign during the second delay period (Fig 5D), illustrating a reversal in the representation of the stimulus as it transitions from being unprioritized to prioritized. This holds true both within the UMI and PMI subspaces.

- *Differential alignment of UMI and PMI subspaces with the decision subspace*: the role of the UMI representation in the 2-back task is to hold information about stimulus *n* in memory that is irrelevant for the impending decision (i.e., when the subject has to make a judgment about stimuli *n + 1* and *n—1*). An important property of this representation, then, is that it should not interfere with that decision. Conversely, the role of the PMI representation is to provide information necessary for the impending decision–it should therefore be able to contribute to that decision. We might thus expect, then, that the activity dimensions that are used to compute the decision should overlap substantially less with the UMI subspace than the PMI subspace. To assess whether this was the case in the trained LSTMs, we used dPCA to extract a decision subspace and then used the metric of Panichello & Buschman [32] to measure the alignment of the UMI and PMI subspaces with this decision subspace. As expected, we find that, whereas the UMI subspace is largely orthogonal to the decision subspace (81.26° ± 2.06° SD), the PMI subspace has substantial overlap with it (35.25° ± 13.25° SD; Fig 7). The UMI subspace was also largely orthogonal to the PMI subspace (84.13° ± 3.91° SD).

## Visualizing EEG activity using dPCA

The EEG data from Wan et al. [13] were markedly noisier than the RNN data: The first two dPCs of the UMI subspace accounted for 69.1% of the total stimulus variance of the trial-averaged data; and the first two dPCs of the PMI subspace accounted for 69.4% of the total stimulus variance of the trial-averaged data.

Regarding the experimental predictions derived from the RNNs in the previous section, we evaluated whether they held in these data:

- *UMI-to-PMI representational reversal*: inspection of the data from a single subject (Fig 6A) shows no signatures of a representational reversal within the first two UMI or PMI dPCs. To confirm this across all subjects, we fit a scalar transformation from the representation in the first half of the first delay period to the representation at every other timestep. The average scalar transformation for each timestep is plotted in Fig 6B. Relative to the UMI subspace, the trajectory of the best-fitting scalar transformation qualitatively matched that from the 60D RNN, increasing across the delay preceding item *n + 1*, then (after holding a constant value across the time interval used to define the subspace) decreasing for the remainder of

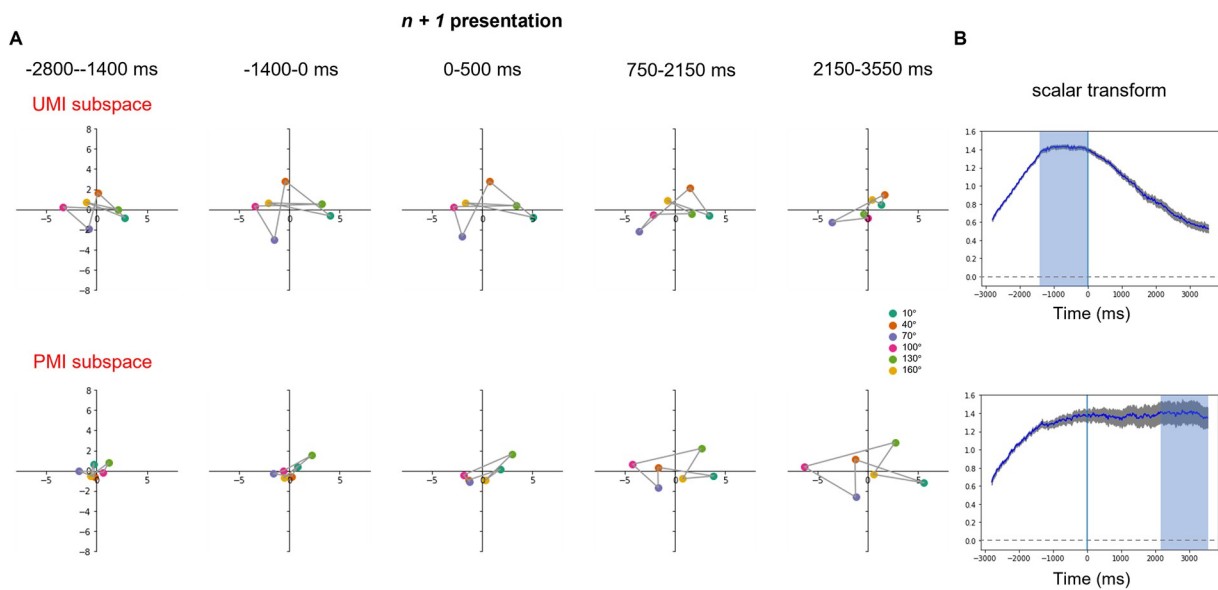

**Fig 6. Stimulus trial-averages of EEG signal projected into UMI and PMI subspaces over a 2-delay time course (-2800ms to 3550ms relative to stimulus _n + 1_ onset).** (A) Results from an example subject. Dot color indicates stimulus _n_'s orientation angle. Data points of adjacent stimulus orientation angles are connected by a gray line. (B) Group-average time course of scalar transform over the course of a trial (N = 42). Blue vertical line indicates the onset of stimulus _n + 1_. Light blue shading shows the time windows that were used to identify the dPCs. The gray shading around the curve shows standard error of the mean.

the trial. Unlike the RNN data, however, the scalar transform never reversed sign (Fig 6B, UMI row). Relative to the PMI subspace, the trajectory for the EEG data started with a steady increase across the delay preceding item _n + 1_, reaching its maximum value while item _n + 1_ was on the screen (i.e., 2 sec. prior to the beginning of the time interval used to define the PMI subspace), then remaining unchanged for the remainder of the trial (Fig 6B, PMI row). This indicates that stimulus representations begin transforming toward their configuration in the PMI subspace, fully achieve it by epoch _n + 1_ (at which time they have UMI status), and then maintain this end-state configuration for the remainder of the trial. This trajectory differs markedly from the 60D RNN, for which the configuration relative to the PMI subspace was unchanging until after delay _1:2_, then rapidly changing across the second half of the trial. Also different from the RNN, the scalar transform for EEG did not reverse sign. These results indicate that representational reversals are not systematically present in the EEG data as they were in the RNN data. Anecdotally, inspection of the EEG data gave the impression of considerably more heterogeneity of representational geometry across subjects (in these first two UMI/PMI dPCs) than we saw across independently trained RNNs.

- *Differential alignment of UMI and PMI subspaces with the decision subspace*: we found that, in the EEG data, the UMI and PMI subspaces were both largely orthogonal with the decision subspace (81.80° ± 6.69° SD and 82.74° ± 8.74° SD, respectively; Fig 7). In other words, we did not observe that the UMI representational subspace had a different geometric relationship to the decision subspace than the PMI subspace. The UMI and PMI subspaces were separated by an angle of 76.87° (*SD* = 12.33°).

## Tracking the disappearance of information from WM using dPCA

As an exploratory endeavor, we have developed an approach using dPCA to track, at the level of an individual neural population (artificial or not), the disappearance of stimulus information

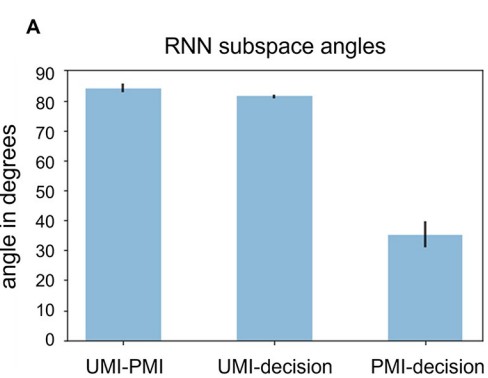
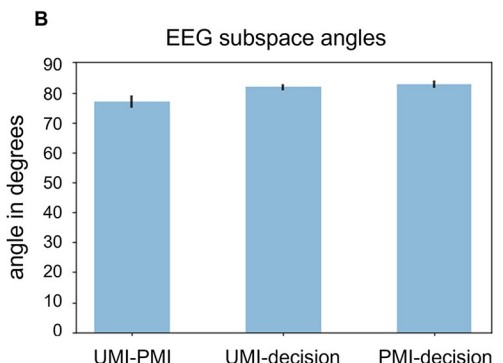

**Fig 7.** Angles between UMI, PMI and decision subspaces for (A) 60D RNN and (B) EEG data. Black bars indicate standard error of the mean.

when it is no longer relevant for a WM task. This was motivated by the observation that, for the three timesteps that follow *n + 2*, stimulus averages projected into the PMI subspace appeared to collapse (Fig C in S1 Text)–what one would expect as information "fades out" of WM at the end of a trial. To quantify this intuition, we start with the three timesteps preceding the onset of stimulus *n* in the time course of stimulus averages projected into the PMI subspace (see Fig C in S1 Text for an example RNN), reasoning that these will not have any information about stimulus *n*. The dispersion of stimulus averages for these timepoints can serve as an empirically derived baseline of discriminability when a stimulus is not in WM. Then, the timesteps immediately following stimulus *n + 2* (when *n* is no longer relevant) can be compared against the pre-trial baseline. To apply this procedure, we would calculate how dispersed the PMI stimuli are from each other at any given timestep and use the resultant metric as a proxy for stimulus discriminability.

We use a bootstrapping procedure in which, for each individual network, we resample with replacement the number of trials from each stimulus condition and perform dPCA on the resampled data as done in the section "*WM-specific dimensionality reduction via dPCA*" (PMI subspace). We then compute the dispersion (i.e., the variance) of the six stimulus trial-averages projected into the PMI subspace, for each timepoint, and repeat the procedure 10,000 times to construct the baseline (null) distribution of dispersion values. To be conservative in rejecting a timepoint as having WM information, we choose, in each iteration, the maximum dispersion value over the values calculated for the 3 timepoints preceding stimulus *n* to construct this distribution. We can then compare any timestep of interest with this distribution to determine whether the dispersion value at the timestep lies beyond the 95th percentile (one-tailed test) of the baseline distribution: if so, we reject the null hypothesis and conclude that this timestep does contain discriminable information about item *n*; if not, we fail to reject the null hypothesis and conclude that we are unable to detect discriminable information about item *n* in its PMI dPC subspace. When applied to the network illustrated in Fig 5C, for example, information about item *n* persisted for one timestep after it was no longer relevant, but then was no longer detectable (Fig C in S1 Text). Unfortunately, this approach cannot be applied to the data from Wan et al. [13], due to their varying lengths of delay. We leave the application of this method to human neural signals for future work.

## Discussion

Results from previous neuroimaging studies have given rise to the idea that representations in working memory (WM) undergo a "priority-based remapping" when they obtain the status of

UMI [11–13], but the mechanism underlying this transformation was unknown. Here, using neural network modeling and dimensionality reduction techniques, we have identified a transition through representational space that may reflect a general solution to the computational problem of needing to hold information in an accessible state (i.e., "in WM") but in a manner that won't influence ongoing behavior. However, noteworthy differences between the transformational dynamics observed with RNNs versus with human EEG suggest important differences in implementational specifics, highlighting important questions for future work.

The 2-back task requires information to transition through three distinct functional states: that of a probe requiring comparison with the mnemonic representation of item *n– 2* and an overt match/nonmatch report; unprioritized (a state that should minimize interference with the concurrent *n– 1* vs. *n + 1* comparison and report); and prioritized (in preparation for its comparison against *n + 2*). PCA of hidden-layer activity of RNNs underwent a smooth rotation through 180˚ of the 2D space defined by the first two PCs. dPCA of RNNs characterized distinct subspaces corresponding to these states, and the trajectories between the states.

The organization of these functional subspaces is reminiscent of recent findings from NHPs performing a retrocuing WM task. Subjects first encoded two stimuli–one above fixation and one below—into WM, then viewed a cue indicating which one to report. Prior to the cue, PCA indicated that the *above* and *below* items were represented in subspaces of neural activity separated by a median angle of 79.1˚. After the cue, the selected item transitioned into a different subspace, and the selected-from-*above* and selected-from-*below* subspaces were closely aligned—separated by only 20.1˚. The authors interpreted this as a transition of the selected item from a representational format that emphasized the distinction between the two items to a "template" format that abstracted over location (no longer a relevant parameter) and facilitated behavioral read-out (specifically, recall; [32]). In our 2-back task, the UMI-to-PMI transition can be understood as the implicit selection of the UMI that occurs after a response is made to item *n + 1*. An important difference between our 2-back task and the retrocuing task of Panichello and Buschman [32], however, is that their task lacked a UMI state. Rather, after the retrocue, there was no possibility that the uncued item would be needed. Nonetheless, in the PFC, a representation of the uncued item persisted, and its uncued subspace was orthogonal to the template subspace. Therefore, one important question for future work is whether, and if so how, the transition to UMI differs from the transition to no-longer-needed (i.e., "dropping" an item from WM).

It is important to note that the RNN modeling that we carried out here is not intended to simulate EEG data, nor the human brain, which has vastly different structural and functional architecture from our RNNs. For example, because of the relative simplicity of the RNN architecture, and the absence of many sources of noise that are characteristic of EEG (e.g., physiological noise, uncontrolled mental activity, measurement noise), the variability and SNR of the two signals differ markedly. This limits what can be interpreted from direct comparisons between the two sets of results. Here we summarize where the two approaches have yielded similar versus dissimilar outcomes, and briefly consider some implications.

## Comparison of RNN vs. EEG results

### Similarities.

- Stimulus representation in both RNN and EEG data went through a priority-based transformation, occupying, in turn, two distinct subspaces (UMI and PMI). This indicates that the UMI was actively represented in both RNN and EEG (c.f., [4–6]). Importantly, it confirms that, at the algorithmic level, prioritization in WM is carried out, at least in part, by an operation of representational transformation.

- The representational trajectories of RNN and EEG data are indicative of an active transformation, and so cannot be accounted for by inhibition (c.f., [17]).

- The angles between UMI and PMI subspaces (RNN: 84˚, EEG: 77˚), and between UMI and decision subspaces (RNN: 81˚, EEG: 82˚) were similar for RNN and EEG. These patterns are consistent with a process that might minimize the influence of the UMI on other concurrent operations, including the retention of the PMI and the processing of the probe.

**Differences.**

- Unlike the RNN data, the EEG data did not show evidence of a sign reversal of the best-fitting scalar transform. It remains to be determined if this reflects a fundamental difference in how the human brain carries out priority-based remapping, or if it may reflect a limitation of extracranial EEG. (E.g., the dynamics of priority-based transformations are different in different brain areas of the NHP [32], but comparable inter-regional differences would be mixed in our whole-scalp EEG data.)

- In the EEG data, the group-average stimulus representation transformed into its final configuration in the PMI subspace earlier than in the RNN data (Figs 5 and 6). (Indeed, human subjects, on average, recoded item *n* into its UMI and PMI configurations simultaneously, and then later prepared for item *n + 2* by collapsing the UMI structure during *delay 2*, whereas the RNNs recoded the item around the time when the priority status of the stimulus changed.) It is also noteworthy that the rate of this transformation was highly variable across individual EEG datasets, but not across RNNs. An important question for future research is whether individual differences in this factor may relate to behavioral performance, as well as whether it is sensitive to such factors as strategy or reward contingency.

- For the EEG data, the angle between PMI and decision subspaces was 83˚, whereas for the RNN it was 35˚. This pattern in the RNN data is consistent with close alignment of these two subspaces that might facilitate comparison of the PMI and the probe. Similar to the point raised previously, future work is needed to determine whether this difference reflects an important difference in decision-making between human and RNNs, or if it is a consequence of poor spatial resolution of the EEG data. (E.g., the effects of selection on WM information are markedly stronger in the PFC than in the visual cortex of NHPs [32].)

## Contributions and limitations of the current work

One important role for the RNN simulations presented here has been to establish the validity and interpretability of our approach with dPCA. This, in turn, allowed us to use dPCA to evaluate neural coding in an EEG data set, including during task epochs for which multivariate methods had failed to find evidence for an active representation of the PMI (Fig 2). This successful application of dPCA to an extant EEG dataset in this study suggests that this approach may also provide novel insights if applied to the data from studies that have previously been interpreted as evidence for activity-silent storage mechanisms [4–6]. The fact that dPCA does not make assumptions about the representational structure of stimuli means that it's possible that it could find evidence for stimulus representation where a model-based approach, such as IEM, has failed. (Indeed, this is what happened with the PMI from the EEG data set in this study–compare Figs 2 with 6.)

It is also important to note that the RNNs we simulated have a simple architecture, with a homogeneous LSTM layer, which is, of course, very different from the brain with its heterogeneous patterns of connectivity between neurons with varied structural and functional

properties. The RNN simulations of Masse et al. [33], employing different cell types and explicitly simulating factors like receptor time constants and presynaptic depletion of neurotransmitter, offer one promising example for developing more biologically plausible models. Also missing from our RNN architecture is an explicit source of control, such as that exerted by prefrontal and posterior parietal circuits in the mammalian brain. Through extensive training, our RNNs gradually learned to adjust their connection weights so as to achieve a high level of performance, but this was only possible because each item presented to the network always followed the same functional trajectory (probe, then UMI, then PMI). A hallmark of WM in the real world is the ability to flexibly respond to unpredictable changes in environmental exigencies. Thus, an important future goal will be to extend the present work to a network with separate modules with different connectivity patterns and governed by different learning rules (e.g., [29,34]), and to a task that requires truly flexible behavior.

Our work complements extant models of attentional prioritization in WM. First, it sheds light on the prioritization mechanisms of a continuous-performance WM task (2-back), a design that has recently received less attention than tasks employing retrocuing. Second, compared with the aforementioned computational accounts [15,17], our use of dPCA provides a data-driven dimensionality reduction approach that does not make assumptions about the representational structure of stimuli. This allows one to examine the unmodeled structure of stimuli in the representational space. Third, our dPCA analyses were applied on a subject-by-subject basis, without assuming that the same representational and/or computational scheme is employed across individuals. Indeed, recent research has shown that representational biases of stimulus features vary among individuals in higher-order brain areas [35]. Therefore, this approach may be helpful for explaining individual differences across many types of cognition.

To conclude, we used ANN simulations to validate the idea, at the level of representational codes, that shifts of priority status trigger the transformation of stimulus representations in WM. Applying dimensionality reduction to LSTM hidden layer activity in RNNs revealed the organization of functionally specific subspaces, and the trajectories between different functional states. This approach translated to EEG data from subjects performing the same task, revealing similarities and differences between human and machine, and highlighting fruitful directions for future research.

## Supporting information

**S1 Text. Fig A.** Example 7-hidden-unit RNN trained with input following the basis function used to build IEMs in Wan et al. [1]. Shown is the 2D visualization of the LSTM hidden layer activity of this RNN. The network architecture and training procedure are identical to the 7D RNNs reported in the main text with the exception that the inputs are not one-hot vectors; instead, they are specified by the IEM basis function: $R = \sin^6(x)$ (e.g., for stimulus #3, input vector is [0.0156, 0.4219, 1, 0.4219, 0.0156, 0]). Note that these results are qualitatively similar to RNNs reported in the main text (Fig 4). **Fig B.** Generating circular input for 60-hidden-unit RNNs. Each point on the circle can be characterized by an angle relative to the easternmost point of the circle. The coordinates of these points within the 2D space on which this circle lives are given by [cos θ, sin θ]. To construct input vectors used in our RNN model, we mapped each stimulus orientation θ to the corresponding point on the circle at 2 * θ. The multiplication by 2 is necessary to match the periodicity of the input vectors to the periodicity of the oriented grating stimuli, which have a period of 180˚ (i.e., the stimulus at θ is equivalent to the stimulus at θ + 180˚). **Fig C.** Empirical test for presence of stimulus information in WM. (A) Time course of stimulus averages projected into the PMI subspace from an example 60D RNN. Data points are colored based on item *n*'s identity. We used the 3 timesteps prior to the presentation

of $n$ to construct a baseline distribution of dispersion values using a bootstrapping procedure. Visually, one can see stimulus information collapsing in the PMI subspace across the three timesteps that follow timestep $n + 2$, colored squares added to identify them for panel B. (B) The baseline distribution of dispersion values, with red dashed line indicating the 95th percentile criterion. Magenta, green and orange lines indicate the dispersion values from timesteps *delay 3*:*1*, *delay 3*:*2*, and *n + 3*, respectively. **Table A.** Cumulative percent variance explained (PEV) by top dPCs of the UMI and PMI subspaces for 7D RNN, 60D RNN and EEG data. The percentages of both stimulus and global variance explained are shown.
(DOCX)

## Acknowledgments

We thank Drs. Yuri Saalmann, Joseph Austerweil, Timothy Rogers, Jacqueline Fulvio, Qing Yu and Mohsen Afrasiabi for helpful discussion and critical feedback.

## Author Contributions

**Conceptualization:** Quan Wan, Bradley R. Postle.

**Data curation:** Quan Wan.

**Formal analysis:** Quan Wan, Jorge A. Menendez.

**Funding acquisition:** Bradley R. Postle.

**Investigation:** Quan Wan, Jorge A. Menendez, Bradley R. Postle.

**Methodology:** Quan Wan, Jorge A. Menendez.

**Project administration:** Quan Wan.

**Software:** Quan Wan, Jorge A. Menendez.

**Supervision:** Bradley R. Postle.

**Validation:** Quan Wan, Jorge A. Menendez.

**Visualization:** Quan Wan, Jorge A. Menendez.

**Writing – original draft:** Quan Wan, Jorge A. Menendez, Bradley R. Postle.

**Writing – review & editing:** Quan Wan, Jorge A. Menendez, Bradley R. Postle.

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
