## [Decision Letter · Decision Letter 0]

11 Jun 2021

Dear Mr. Wan,

Thank you very much for submitting your manuscript "Rotational remapping between differently prioritized representations in visual working memory" for consideration at PLOS Computational Biology.

As with all papers reviewed by the journal, your manuscript was reviewed by members of the editorial board and by several independent reviewers. In light of the reviews (below this email), we would like to invite the resubmission of a significantly-revised version that takes into account the reviewers' comments.

As you will see below, all four reviewers found the work interesting and of high importance. However, there was also a consensus opinion that the description of the model should be expanded and that a broader range of parameters might provide deeper insight the mechanisms. Similarly, while all of the reviewers found the comparison of the EEG and RNN results impressive, they were hoping for a more in-depth discussion, particularly of any differences that were observed. Finally, a few reviewers raised interesting points about the RI statistic, its definition, and comparison to other transformations.

We cannot make any decision about publication until we have seen the revised manuscript and your response to the reviewers' comments. Your revised manuscript is also likely to be sent to reviewers for further evaluation.

Sincerely,

Tim Buschman

Guest Editor

PLOS Computational Biology

Thomas Serre

Deputy Editor

PLOS Computational Biology

Reviewer's Responses to Questions

**Comments to the Authors:**

Reviewer #1: In this manuscript, Wan and colleagues address a timely question that could be of interest to the readership of Plos Computational Biology. However, I find the initial framing of the paper confusing (and long) and I think the analyses seem preliminary. Below I provide a few suggestions that the authors might consider to improve the paper.

Framing

There have been intense debates about the neural basis of prioritized vs unprioritized working memory maintenance, with diverse evidence pointing to storage in different brain areas, reversed neural codes (the hypothesis addressed here) or ‘activity-silent’ mechanisms.

It is important to stress that some of these hypotheses are mutually exclusive, in contrast to what the authors claim.

For example, if a memory is stored in an *active* reverse code, does it make sense to talk about a *silent* code? If it is reversed, it is not silent, since silent is the opposite of active. Of course, there are many biological processes co-occurring in the brain that could be considered “silent” (i.e. not based on spiking activity), such as forms of plasticity, channel adaptation, etc. An active reverse code could then coexist - and most likely it does - with a silent process. However, the likely possibility that an active code might coexist with silent mechanisms is not evidence for WM maintenance through silent mechanisms. This confusion is pervasive in the field and this manuscript somehow helps perpetuate it. The work described here is a good proof of concept that a reverse code does not need any sort of “silent” mechanism and thus the paper could have been framed as such. Indeed, without touching the results section, one could frame the paper as disqualifying the storage of UMI in activity-silent codes, similar to what has been done for reactivations seen in fMRI: "Restoration of fMRI Decodability Does Not Imply Latent Working Memory States" by Schneegans & Bays in Journal of Cognitive Neuroscience (2017) 29 (12): 1977–1994

In general, I think an explicit discussion of this apparent contradiction would be valuable to clarify the current confusion. For example, how come the UMI decoding in Rose et al was at chance levels, but here, a below-chance code is found empirically and modeled with RNNs without short-term plasticity? Isn’t the proposed RNN model here in contradiction with Rose et al, whereas the authors say: “UMIs may be maintained as “activity-silent” traces encoded in synaptic weights”?

Instead, the authors gloss over the controversy and ignore papers that consistently show (including their own EEG work and now modeling) that UMI are stored in active codes.

Analyses

The authors show interesting rotations in the RRN model, both by quantifying it and visualizing it in a 2-D principal component space. They also quantify similar rotations in EEG data; however, without showing the respective visualization (why?).

I found the explanations/interpretations in the captions and the text (just 2 pages for results!) unsatisfactory.

For example, Figure 4 A and B look completely different, while the purpose of this study is to highlight their similarities. Improving this figure would be key to send the main message across for quick readers.

Figure 2 is copied from the original EEG study, but never replicated in the RRN. Generating an equivalent figure for simulated data would also make the main point more convincing. On the other hand, Figure 3, which is very difficult to follow, is never replicated in the EEG data.

It is not clear how the reverse code emerges in EEG (I guess training on AMI but testing on UMI?). The paper is motivated by and centered around the finding of inverted decoding. Yet, it is not clear how rotations can lead to such inverted decoding. I guess inverted decoding in EEG signals would mean that AMI are stored in some area(s) while the UMI are stored in other(s)? At a minimum, it should be shown in what conditions the rotations and inverted encoding do not emerge. For example, would increasing the number of neurons (now 7!) still bring about these rotations? Would one get the same rotations if the RNN were trained on an abstraction of the Rose task? If so, what does that tell us of the activity-silent interpretation of that paper? It is thus not very clear what was learned with this novel modeling work. The main purpose of training RNNs is to go beyond what is done in the data and propose mechanisms for an otherwise descriptive finding. The simple observation that similar dynamics occur both in the data and network is unsatisfactory to me and I guess as well for a general computational readership.

General comments, as I re-read the paper

The introduction and discussion has excessive description of previous experiments, even defining variables that are never used throughout the paper:

“the basis function parameters for memory strength (φ), gain (γ), receptive field width (μ), and receptive field centers (δ) were varied. “

On the other hand, the results were too short. The authors could consider reducing the introduction and discussion and clarifying the results further.

Overall, I feel there are too many figures to show one simple - however interesting - point, which is that RNNs show rotations, as seen in EEG data. For example, figure 6 can be substituted by one short sentence. I had a hard time jumping from figure to figure while most figures could be collapsed into one figure.

Why so few networks (and neurons!)? Figures would look much more convincing if many more networks would populate Figure 5, for example.

The technical details about the RNN are not sufficiently clear. For example, the authors only used 7 neurons but an input of 6 dimensions. How much do the results depend on these choices? I would consider using bigger networks and including a diagram of the network architecture as well as the task used for training in Figure 1.

The authors discussed the results of Libby and Buschman (2021), here mice passively listen to auditory stimuli, to great extent. There is no mention of another work from the same lab that directly investigated the neural mechanisms of UMI vs AMI. This study seems extremely relevant to the modelling work produced here and I would recommend discussing it.

Reviewer #2: Wan et al re-analyse EEG signals during a working memory 2-back task, and compare the EEG decoding with the hidden layer of a LSTM RNN trained to perform the task. This is a very hot topic and an interesting new approach. Notably they introduce a new metric, the "rotation index" which helps to quantify representational rotations in dimensionality-reduced data. The rotation may be of crucial importance in protecting information in WM from interference.

I enjoyed reading this manuscript and it has a great deal of novelty. I have mainly minor clarificatory points that I think might improve the paper.

1. The procrustes solution for a rotation assumes the mapping is a rotation rather than a scaling/negation. In other words, if the data showed a flip, this fitting would "force" it into a rotation. Note that a 180 degree rotation is the same thing as a "flip" of both the projected components (i.e. reflection on both axes), so it is possible that this kind of "flip" (scaling by negative value) might fit reasonably well. Importantly, a situation where there was *no* rotation but just a scaling, would be well-fitted by a rotation of zero. I may have misunderstood their permutation method, but it seems that a rotation of zero would give a significant rotation index?

For these reasons it would be nice if the authors compared the rotation R with a simple scaling (or even a combination of rotation*scaling). After all, this is the alternative hypothesis they flag up in the intro.

2. p5 para1 - "Notably, these “flipped” (or “opposite”, van Loon et al., 2018) states are not less active relative to the PMI representations, rather, they are different."

This sentence in brackets is unclear. In what way were they "different"? It seems the authors are suggsting why it was not simply a polarity change, but that doesn't seem with the IEM reading out a flipped response.

3. p12 first para: "Namely, we added an extra stimulus event following the stimulus n trial. We only plotted..." This sentence was hard to follow: why was an extra event added, what was the stimulus? What is an 'overlapping' stimulus event and why are those trials not plotted? Maybe it can be illustrated on fig1.

4. Might be helpful to adopt uniform naming of timpoints and reflect them in Fig1 e.g. I assume "delay 1:1-2" is the frame after "n" in Fig1? Fig 2 may need a bit more explanation, colormap legend, and meaning of "response" y-axes. Would a schematic, rather than this complex figure, be better?

5. p16 line 1: "subspaces might be overlapping" - unclear what "overlapping" means for a pair of linear subspaces. Do the authors mean they may share one of their two axes? intersect?

Similarly, p21 second para, "confirming that these two subspaces overlap" - I am not sure of the logic why the presence of rotations in both directions confirms the spaces overlap. It may be that there are two orthogonal spaces, both of which have rotations? [ Perhaps this can be determined from the inner products of the columns of the two W matrices. ]

7. I really like the clever definition of the rotation index, which will be useful to future research. The authors could highligh this as part of the result/impact of this paper. Fig 4 could even visualise the rotated UMI against the PMI, or rotated PMI against the UMI, to show how the degree of alignment determines this index.

6. There are three interesting ways the model deviates from the data. a) The data shows a strong 2-dimensional code (second PC is useful) for both the UMI and PMI. This doesn't seem to match the model. The authors should discuss why this might be. b) It is interesting that the PCA fails for EEG data but not for the RNN - could the authors discuss why this might be? c) are the EEG rotation angles significantly larger than the RNN's? I think these differences are of potential interest.

7. Fig 3: do the axes show y_UMI or y_PMI?

I am guessing UMI because there is better separation for n+1. How was the "stimulus coding axis" determined? Panel S4A was helpful in understanding this and I wondered if it could be incorporated into fig4 somehow. I think figure 6 is redundant given fig 5?

8. p24: dPCA "does not make assumptions

about the representational structure of stimuli" -- it assumes the stimuli are represented linearly among neurons.

The authors should be congratulated on a very interesting study.

Reviewer #3: Thanks for giving me the opportunity to contribute to the review process. This is a very interesting, novel and well written study on a candidate mechanism underlying the retention of unattended memory information (UMI). While there is a surge of evidence that the UMI reconstructions (i.e., colors, orientations etc.) are flipped relative to the attended (PMI) ones, it is not known how this is achieved at the mechanistic level. The authors provide a timely and important answer to this question. They model WM performance on a 2-back task using RNN-based LSTM models. The dynamic evolution of the hidden layers is inspected in detail using dimension reduction methods (PCA and dPCA). They convincingly show that a rotational remapping process characterizes the UMI-to-PMI state changes as index by RI. Interestingly, a similar rotational remapping was observed when dPCA was applied to EEG data (from a 2-back task). Overall, I believe this study is an important contribution to the field and has the potential of further inspiring new theoretical and methodological insights. Below I have listed some (mainly) clarification questions and (minor) comments that need to be addressed before publication.

The authors provide extensive details about PCA, dPCA and RI. Information about the RNN-LTSM is however sparse. Given its central role in the manuscript, the authors should provide more details about the model in the manuscript. While readers will have access to the materials on osf and where most of the questions below will find their answers, I could not consult the following information:

- Please define the RNN/LSTM more precisely, as it is currently difficult to understand how it is constructed. By definition, the application of LSTM implies there were forget units, remember units, sigmoids, and tanh functions in the hidden layer. If so, or otherwise, please provide this information in the methods.

- As far as I can interpret the model, there are multiple inputs, that are sequentially dependent, thanks to the recurrent nature of the network in the hidden layer. Given that the trial event is a concatenation of n, n+1 and n+2 along with the 2 delay periods each, it is not clear whether there is only 1 output (many-to-one), that of the n+2 target, or multiple outputs, that of each individual event: n, n+1 and n+2 along with the 2 delay periods (many-to-many). A graph depicting the model architecture could be helpful.

- Whether and how hyperparameters were optimized: any dropout or regularization?

- Whether the input states of a non-match and a delay period -that are both set to [0 0 0 0 0 0]- are functionally the same?

- Ten RNNs were used (N=10). Please provide information on how these 10 RNNs vary. Do they vary in terms of their architecture, parameters, or train/test instances?

- Please explain why 2 delay periods were installed (does this reflect the time points before and after the mask in 2-back task? Or does this number relate to the length/duration of the interval?)

The discussion provides plausible interpretations to the findings and draws connections to the literature (e.g., Libby & Buschman, 2021). Additionally, the limitations of the current study are addressed along with interesting future directions. I believe that the authors should also discuss potential reasons why rotational remapping was only revealed in the subspaces obtained from the dPCA alone and not with generic PCA. This is an empirical question that is probably beyond the scope of the current manuscript. My concern is that dPCA, which maximizes between-group (PMI vs UMI) variability and minimizes within-group variability, orthogonalizes the high dimensional feature space relative to the PMI/UMI conditions separately. Consequently, the subspaces that are optimally responsive to a specific condition may result in a flip or rotation in the other condition (potentially induced by the dPCA orthogonalization?).

Finally, does rotational remapping predict any behavior? The correlation between RI and ACC/d’ cannot be achieved due to ceiling performance and this is something that the authors have addressed in the discussion. However, I was wondering whether the authors have tried to look at any relation between RI and speed? The reason to establish such a link is to bridge the gap that exists now between the model and human data. The LSTM nicely models the 2-back performance, and the rotational remapping is derived from the hidden units’ activity in PCA and dPCA subspaces. So, the model’s inner workings are inherently coupled to performance. In analogy to this, any link between the RI and speed would be interesting to further support the functional role of rotational remapping.

Minor:

- It would be better to spell out LSTM (for the first time) for the sake of clarity for a broad readership. Same for ANN in the discussion.

- The following sentence is related to the RNN, but the referenced figure is about the IEM data. Is this correct? (p13) ‘As with the RNN data, after excluding the first two stimuli from each block there were 126 stimulus events and hence 125 trials per block (Figure 2).’

- RNN training and testing: “The activity timeseries of the LSTM hidden layer units from all 3200 trials (16 trials x 200 sequences) in the training data set were extracted for subsequent analyse” (P11) Is this correct? Were subsequent analyses and visualizations performed on the training set or on the testing set? For the sake of generalizability, the testing set should be considered. And to clarify, was the number of extracted data points not = 16 trials x 200 sequences x 3 time points (= stimulus + delay 1 + delay 2)?

- The idea of ‘overlapping stimulus events’ is not very clear. It could be me but I have difficulties in understanding the following: ‘(e.g., the n-plus-n + 1 trial and the n + 2-plus-n + 3 trial were plotted but not the n + 1-plus-n + 2 trial; consequently, a total of 1600 trials were plotted).’

Reviewer #4: In their paper, Wan, Menendez, and Postle present rotational remapping as a plausible account for how unprioritized working memory items (UMIs) are neurally represented. Prior work has shown that UMIs may be represented in activity-silent traces and/or as inverted or suppressed representations. This paper takes the approach of using RNNs to try to gain insight into the mechanisms underlying UMI-to-PMI representational transformations. RNNs were trained to perform a 2-back WM task similar to the EEG task from Wan et al. (2020). The authors first use PCA to visualize the hidden layer of the RNNs, showing that the UMI (the currently task-irrelevant item in WM that will become task-relevant on the next trial) is represented similarly but rotated compared to the PMI (the currently relevant, prioritized WM item). Moreover, the representation seems to gradually rotate over time in transforming from the UMI to PMI. They then use a demixed PCA analysis to quantify a rotational index. Similar analyses were performed on the EEG data from Wan et al. (2020), again showing UMI representations rotated relative to PMI. The authors conclude that rotational remapping is a candidate computation for dynamic WM prioritization, and shows promise for artificial neural networks to aid mechanistic interpretations for human cognition.

Overall, the demonstration of rotationally remapped UMIs in both the RNN and EEG data is very interesting! The topic of how WM items of different priorities are represented neurally is important and timely, and I read this paper with great interest. The authors did well in discussing how their findings are bridged to other work, and in raising important theoretical questions in the discussion. That said, I found several aspects of the methods and interpretation confusing and/or lacking in detail, and was left with an overall sense of not really knowing quite what these results mean in terms of neural representations. If the authors are able to clarify and provide a clearer link between the RNN and EEG data, I think the ideas in this paper would make a nice contribution to the literature.

(Also, note that I am not an expert in RNN methods, so I did not evaluate the details of that aspect of the manuscript.)

Main concerns:

1) What exactly is meant by “rotational remapping”? Is it the idea that the UMI is a rotated version of the PMI? Or is it referring to the dynamic mechanism by which the UMI transforms into the PMI via rotation over time? This is important because the RNN shows both, but the EEG data are only showing the first. Which is the main contribution of the paper?

2) I could clearly see the rotational remapping in the RNN representations, but I kept finding myself getting stuck trying to translate that intuitively into what that means in terms of how the brain represents the stimuli.

a. Is the idea that the brain is actually performing computational steps similar to dPCA? If so, what would the RNN hidden layer units reflect in the brain?

b. I also had difficulty trying to intuitively track a single stimulus through this transformation and how it would result in the match/no-match outcome. For example, at time n, a “blue” stimulus is represented in the right side of this representational space. Then it rotates such that that stimulus is now located towards the top of the space during the UMI period 1:2. Then it continues rotating so that it is in the upper left during PMI period 2:2. And then at time n+2 it’s rotated back to the “decision”-aligned axis. But it’s now in the opposite direction (left side) of the representational space. How would this representation of the stimulus n at time n+2 be compared to the incoming n+2 stimulus to determine a match, and why are the matches clustered in the center of that representational space? Some sort of cartoon walk-through or intuitive explanation would be useful here.

c. Is the rotational remapping related to the continuous/circular nature of the orientation space? I know that the RNN was not fed orientations per se, just 6 labeled features, but what information did the RNN have about the relationship between those features? Is it similar to orientation in that feature 2 is more similar to feature 3 than feature 4? (If not, why did it produce a representation showing a seemingly continuous structure across the stimulus types?) Is it circular space like orientation? On an intuitive level, I feel like a rotational remapping of orientation space kind of makes sense, and would produce the inverted IEMs. But if the UMI and PMI were flowers and cows, what would this rotational remapping look like?

3) The EEG data seem to only partially confirm the RNN data. A main feature of the RNN results was the timecourse analysis and gradual rotating of the data. Could a timecourse analysis be done on the EEG data? At minimum the delays could be split into 2 timepoints each to mimic the RNN analysis (which might actually clean up the EEG data some if there was continuous rotation). But the EEG data could lend themselves to something even cooler: a timepoint by timepoint analysis with higher temporal resolution, plotting the rotation angle over time.

4) Also, what does it mean that the EEG data showed rotation with dPCA but not PCA?

5) The rotation index (RI) measure:

a. More clarity is needed in describing this measure. I did not follow if lower RI values mean more rotation or a better-fitting rotational structure. For example, if the representation rotates 180° but this rotation is very noisy, would that have a higher or lower RI compared to a representation that rotates 10° but the rotation is extremely precise?

b. The authors used permutation testing to determine significance, but it appears the shuffling of labels was done after the dPCA procedure. I’m not sure this is valid. Wouldn’t it be better to shuffle the data labels before the dPCA to obtain a true null distribution?

c. On p.17 it says a p<.05 would indicate a “pure rotation”. Is that a valid statement? Or is it simply indicating a rotation greater than expected by chance?

Minor points

6) A simple figure or schematic to visually show the reader the architecture of the RNN would be much appreciated. Also some explanation of how certain parameters were decided (e.g. why 10 RNNs, 7 units, and different numbers of trials & blocks vs human EEG data?)

7) As someone not an expert in RNNs, it struck me the near-perfect accuracy of the RNN at performing this task. Is that typical? If the human participants are performing the task at a considerably lower accuracy, what does that mean for the comparison? Related point: if only highly accurate RNNs were included in the analysis, why not perform the EEG analysis with only correct trials included?

8) Figure 2 I had a bit of trouble following the details. What is the scale/units on the color bar? Why is this different from the scale of the y-axis on the reconstructions in the right plots? Also, is channel 0 centered on stimulus “n” at all timepoints? (If so, why isn’t there a strong reconstruction of the stimulus during the initial presentation of “n”?)

9) In Figure 3, the black dashed line illustrates the “stimulus coding axis” and the blue dotted line depicts the “decision-based structure”. These are very important terms as they are used for interpreting rotational representations (e.g., the decision-based structure becomes perpendicular to the stimulus coding axis) but these terms do not seem to be explicitly defined in the text. How were these axes determined? Computationally, or by eyeballing a best fit on the plots? (Also, what’s the significance of the band-like representational structure?)

10) For Figure 4B (the EEG data), it’s harder to see the patterns. A few suggestions:

a. Consider using color coded dots. I understand grayscale was used so as not to confuse the orientations with the arbitrary RNN conditions, but then it would be more intuitive to color code 4B and have 4A grayscale. In any case, the gray legend is hard to see and compare, it doesn’t capture the circular nature, and the connected lines didn’t seem to help.

b. Perhaps it would also help to have a third column to visually display the rotated UMI (based on the rot.angle), so readers can directly compare PMI and “rotated UMI”?

c. It would also be helpful to see the plot of the dPCA subspaces next to the means, as in Figure S4.

11) The definitions of stimulus and global variance were not clear.

12) The linked OSF page is currently empty (but I appreciate the authors making their data available for open-access!).

**Have the authors made all data and (if applicable) computational code underlying the findings in their manuscript fully available?**

Reviewer #1: **No: **There is a link to the data, but the repository is empty.

Reviewer #2: Yes

Reviewer #3: **No: **The OSF link was available in the manuscript but I could not find the materials (I am not sure if this is due to me, access failure)

Reviewer #4: **No: **OSF page is listed but currently empty

PLOS authors have the option to publish the peer review history of their article (what does this mean?). If published, this will include your full peer review and any attached files.

Reviewer #1: No

Reviewer #2: No

Reviewer #3: No

Reviewer #4: No
---

## [Decision Letter · Decision Letter 1]

24 Feb 2022

Dear Mr. Wan,

Thank you very much for submitting your manuscript "Priority-based transformations of stimulus representation in visual working memory" for consideration at PLOS Computational Biology. As with all papers reviewed by the journal, your manuscript was reviewed by members of the editorial board and by several independent reviewers. The reviewers appreciated the attention to an important topic. Based on the reviews, we are likely to accept this manuscript for publication, providing that you modify the manuscript according to the review recommendations.

All of the reviewers feel the revised manuscript is significantly improved. However, as detailed in their comments, a few minor concerns remain. In particular, Reviewers 1, 2 and 4 all highlight the importance of clarifying the relationship between the EEG data and the RNNs. Reviewers 1 and 2 suggest clarifying how the nature of EEG recordings may affect the interpretation of the results. Finally, Reviewer 4 notes a difference in the exact task performed by the RNNs and subjects and whether aligning these more closely may change the results.

Sincerely,

Tim Buschman

Guest Editor

PLOS Computational Biology

Thomas Serre

Deputy Editor

PLOS Computational Biology

[LINK]

All of the reviewers feel the revised manuscript is significantly improved. However, as detailed in their comments, a few minor concerns remain. In particular, Reviewers 1, 2 and 4 all highlight the importance of clarifying the relationship between the EEG data and the RNNs. Reviewers 1 and 2 suggest clarifying how the nature of EEG recordings may affect the interpretation of the results. Finally, Reviewer 4 notes a difference in the exact task performed by the RNNs and subjects and whether aligning these more closely may change the results.

Reviewer's Responses to Questions

**Comments to the Authors:**

Reviewer #1: While I think the paper has improved after the revision, several of my concerns were not properly addressed, as I detail below (not sorted by relevance).

1. To be clear, I did not suggest the authors reframe the paper. My point was that the previous version of the manuscript touched explicitly on the current debate of active vs silent WM right from the outset, but then did not position their results within this debate. In the current manuscript, the authors say ‘the present report does not relate to this question’, but this is not true. This report aims at clarifying the mechanisms of UMI storage, precisely the aforementioned debate. That being said, I think the manuscript in the current form is clearer in this regard. I appreciate that the authors now acknowledge in the introduction that they are offering an alternative explanation to the empirical finding of inverted encoding, but I think the sentence ‘the present report does not relate to this question’ should be removed.

2. My remark about having ignored the apparent contradiction between the findings of Rose et al. and Wan et al. remains an issue. Note that this (and other concerns) should be addressed directly in the manuscript, not in the rebuttal.

3. My remark about the inverted encoding potentially originating from decoding from different areas/sensors was not addressed. This is a hypothesis that can be tested directly in the data at hand, without the need to collect more data or train new RNNs.

4. Why include networks with 7 units after all? The author's justification (default pytorch configuration) is unsatisfactory. Removing these networks from the manuscript would substantially improve the readability of the manuscript.

5. The main claim of this report is the link between data and the RNN. I appreciate the transparency in the abstract about this link (or the lack thereof), but I still think figure 5 and 6 (perhaps the main results?) are rather obscure and don’t transmit the idea that data and RNN are comparable - quite the opposite?

5. In the new version of the manuscript, I was a surprised by the sentence ‘Thus, although MVPA and IEM produced the results that gave rise to the priority-based remapping hypothesis, [they] are poorly suited to evaluating it, because they don’t permit direct measures of neural representation’, because the limitation is not in the decoding method but the nature of the EEG data.

Reviewer #2: Wan et al have substantially revised their paper. The rotational mappinng no longer features in the EEG data, and instead a scale re-mapping is used. The story is different but remains highly topical. Many parts of the text have been clarified.

My only suggestion is to make the abstract more explicit about the main finding. I think their interpretation is that RNNs effectively perform smooth rotation, preserving information, while EEG signals could suggest a scaling down / collapsing.

This would help highlight the importance of their nice results. Just to interpret further, though not wanting to be too controversial, I wonder if the new analysis suggests that information that is "out of mind" is "out of sight", at least from EEG.

Reviewer #3: Thank you for your responses to my review. The authors have answered all my questions and responded quite well to my comments. I find this revision along with the very interesting additions (eg scaling, dispersion, UMI-PMI-decision angles) satisfactory.

Reviewer #4: This manuscript is a revised version of one I previously reviewed (Reviewer 4). The authors have clearly put a lot of work into this revision, which is impressive and commendable. The manuscript now includes much clearer descriptions of the methods and results, and the conclusions seem more in line with the data. So in that sense, the authors have addressed most of my specific concerns and I find the manuscript much improved.

However, the resulting changes to the manuscript now paint a picture where the RNN results are neat and interesting, but they really don’t match at all with the EEG results. Again, this is handled well in the revision: the results are organized into clear comparisons/hypotheses, and the discussion doesn’t over-extend the findings. But now I’m struggling with what it means if an RNN shows something but the neural data do not, and what we can/should take away from that.

A few specific issues I find myself getting hung up on:

- The manuscript now clarifies that the RNN was trained on 6 discrete features with arbitrary relationships, and this differs from the continuous circular relationship between the 6 orientations of the EEG experiment. But, why? This seems a fundamental difference between the two. I get that the authors wanted to have the RNN results not be idiosyncratic to the circularly related stimuli, but that only makes sense if they found similar results across the RNN and EEG. Given that the results are now quite different, I’m a bit lost in how much stock to put in the RNN findings. It feels like there should be a comparison model where the 6 features of the RNN are coded circularly and continuously, to see if these representational transformations persist in that context.

- I may have missed this in the authors response, but why do the EEG data in the revised manuscript (Fig 6) look so different from the initial manuscript (Fig 4)? In the original manuscript, the EEG projections seemed to transform/rotate substantially between the UMI and PMI representations, but now that doesn’t really seem to happen. As described in the revision, there’s no suggestion of a reversal and the angle separating them is much smaller. Can one even conclude from the EEG data that there’s a transformation / progression in representational state from UMI to PMI? I feel like I’m missing something.

So overall I’m not sure what to think about this manuscript. The authors have put a great deal of work into an improved revision, and I really do want to acknowledge that, despite my reservations.

**Have the authors made all data and (if applicable) computational code underlying the findings in their manuscript fully available?**

Reviewer #1: Yes

Reviewer #2: Yes

Reviewer #3: Yes

Reviewer #4: None

PLOS authors have the option to publish the peer review history of their article (what does this mean?). If published, this will include your full peer review and any attached files.

Reviewer #1: No

Reviewer #2: **Yes: **Sanjay G Manohar

Reviewer #3: No

Reviewer #4: No

Figure Files:

Data Requirements:

Reproducibility:

References:

---

## [Editor Report · Decision Letter 2]

3 May 2022

Dear Mr. Wan,

Thank you very much for submitting your manuscript "Priority-based transformations of stimulus representation in visual working memory" for consideration at PLOS Computational Biology. I apologize for the long delay in rendering a decision. As noted below, we have been extensively discussing the manuscript. 

After lengthy discussion among the editors, we are excited to move forward with publishing the manuscript but feel that the manuscript should be further revised to magnify the discussion of the differences between the model and experimental data. As you know, neural network models are typically viewed as providing insight into the computations of the brain. For this reason, when comparing models and the brain, most manuscripts focus on the ways in which the brain and models are consistent. In contrast, the current manuscript highlights differences between neural recordings and commonly used computational models of working memory tasks. During the last round several reviewers raised concerns about the interpretability of the mismatch between models and the brain.

However, we agree with the authors that there is value in discussing how models and neural networks differ. We feel that these discrepancies can provide just as much insight into the mechanisms and also play an important role in avoiding over-interpretation of computational models. Therefore, we hope you would be willing to further revise the Introduction and Discussion of the manuscript to place greater emphasis on a) the general motivation that led you to compare computational models and the brain and b) discussing and interpreting the differences between the model and neural data.

Sincerely,

Tim Buschman

Guest Editor

PLOS Computational Biology

Thomas Serre

Deputy Editor

PLOS Computational Biology

[LINK]

I apologize for the long delay in responding. After lengthy discussion among the editors, we are excited to move forward with publishing the manuscript but feel that the manuscript should be further revised to magnify the discussion of the differences between the model and experimental data. This is reflecting the feeling from several reviewers in the last round.

As you know, neural network models are typically viewed as providing insight into the computations of the brain. For this reason, when comparing models and the brain, most manuscripts focus on the ways in which the brain and models are consistent. However, we agree with the authors that there is value in also discussing how models and neural networks differ. We feel that these discrepancies can provide just as much insight into the mechanisms and also play an important role in avoiding over-interpretation of computational models. Therefore, we hope you would be willing to further revise the Introduction and Discussion of the manuscript to place greater emphasis on a) the general motivation that led you to compare computational models and the brain and b) discussing and interpreting the differences between the model and neural data.

Figure Files:

Data Requirements:

Reproducibility:

References:

---

## [Editor Report · Decision Letter 3]

12 May 2022

Dear Mr. Wan,

We are pleased to inform you that your manuscript 'Priority-based transformations of stimulus representation in visual working memory' has been provisionally accepted for publication in PLOS Computational Biology.

Best regards,

Tim Buschman

Guest Editor

PLOS Computational Biology

Thomas Serre

Deputy Editor

PLOS Computational Biology

---

## [Editor Report · Acceptance letter]

27 May 2022

PCOMPBIOL-D-21-00800R3 

Priority-based transformations of stimulus representation in visual working memory

Dear Dr Wan,

I am pleased to inform you that your manuscript has been formally accepted for publication in PLOS Computational Biology. Your manuscript is now with our production department and you will be notified of the publication date in due course.

With kind regards,

Zsofia Freund
